# How does the ice sheet surface mass balance relate to snowfall? Insights from a ground-based precipitation radar in East Antarctica

Niels Souverijns[1], Alexandra Gossart[1], Irina V. Gorodetskaya[2], Stef Lhermitte[3], Alexander Mangold[4], Quentin Laffineur[4], Andy Delcloo[4], and Nicole P.M. van Lipzig[1]

[1]Department of Earth and Environmental Sciences, KU Leuven, Belgium
[2]CESAM - Centre for Environmental and Marine Studies, Department of Physics, University of Aveiro, Portugal
[3]Department of Geoscience and Remote Sensing, Delft University of Technology, The Netherlands
[4]Observations Department, Royal Meteorological Institute of Belgium, Uccle, Belgium

*Correspondence to:* Niels Souverijns (niels.souverijns@kuleuven.be)

**Abstract.** Local surface mass balance (SMB) measurements are crucial for understanding changes in the total mass of the Antarctic Ice Sheet, including its contribution to sea level rise. Despite continuous attempts to decipher mechanisms controlling the local and regional SMB, a clear understanding of the separate components is still lacking, while snowfall measurements are almost absent. In this study, the different terms of the SMB are quantified at the Princess Elisabeth (PE) station in Dronning Maud Land, East Antarctica. Furthermore, the relation between snowfall and accumulation at the surface is investigated. To achieve this, a unique collocated set of ground-based and in-situ remote sensing instrumentation (Micro Rain Radar, ceilometer, Automatic Weather Station, among others) was set up operating for a time period of 37 months. Snowfall originates mainly from moist and warm air advected from lower latitudes associated with cyclone activity. However, snowfall events are not always associated with accumulation. During 38 % of the observed snowfall cases, the freshly-fallen snow is ablated by the wind during the course of the event. Generally, snow storms of longer duration and larger spatial extent have a higher chance to attain accumulation at the local scale, while shorter events usually result in ablation (on average 17 and 12 hours respectively). A large part of the accumulation at the station takes place when preceding snowfall events were occurring in synoptic upstream areas. This fresh snow is easily picked up and transported in shallow drifting snow layers over tens of kilometers, even when wind speeds are relatively low ($< 7$ ms$^{-1}$). Ablation events are mainly related to katabatic winds originating from the Antarctic plateau and the mountain ranges in the south. These dry winds are able to remove snow and lead to a decrease in the local SMB. This work highlights that the local SMB is strongly influenced by synoptic upstream conditions.

## 1 Introduction

The Antarctic Ice Sheet (AIS), being currently the largest ice body on earth, is an important regulator of present and future sea level and the global water cycle (Vaughan et al., 2013; Previdi and Polvani, 2016). In order to asses its contribution to sea level rise, understanding the surface mass balance (SMB) of the AIS is of crucial importance. Climate models (potentially coupled to an ocean or ice sheet model) play an important role in understanding and quantifying the contribution of the AIS to sea level (rise) (Gregory and Huybrechts, 2006; Rignot et al., 2011; Ligtenberg et al., 2013; DeConto and Pollard, 2016). Yet, despite

their importance, they simplify the different components of the SMB. It has been noted by several authors that in order to fully understand the impact of the AIS on future sea level rise, information on the individual components of the present-day AIS SMB, including snowfall measurements, is indispensable (van Lipzig et al., 2002; Rignot et al., 2011; Shepherd et al., 2012; Agosta et al., 2013; Gorodetskaya et al., 2015; Lenaerts et al., 2016; Grazioli et al., 2017a, b; Souverijns et al., 2017). A lack of observations of the present behaviour of these different components prevents the validation of climate models. As such, most studies rely on stake and other measurements that register only the total change in snow height (Vaughan et al., 1999; van Lipzig et al., 2004; Genthon et al., 2005; Magand et al., 2007; Eisen et al., 2008; Agosta et al., 2012; Favier et al., 2013; Wang et al., 2016; Favier et al., 2017). Gorodetskaya et al. (2015) quantified the different terms of the local SMB in a systematic way for the Princess Elisabeth (PE) station and determined snowfall from radar measurements. In this study we focus on the relation and interactions between snowfall and accumulation in order to understand and assess the local SMB components in more detail.

The local SMB is influenced by several processes and can be expressed as the sum of snowfall (S), surface sublimation ($SU_s$), drifting snow sublimation ($SU_{ds}$), surface melt (ME) and wind-induced accumulation or ablation by drifting and blowing snow ($ER_{ds}$) (van den Broeke et al., 2004):

$$SMB = S + SU_s + SU_{ds} + ME + ER_{ds} \qquad (1)$$

Although previous studies proposed a variety of techniques to calculate the local SMB, most are confined to measuring the sum of all components (net height change at the surface) using stake measurements and ice cores (Vaughan et al., 1999; Rotschky et al., 2007; Favier et al., 2013; Wang et al., 2016; Thomas et al., 2017) or to one-dimensional snow models (Groot Zwaaftink et al., 2013). Generally, the separate measurement of any of the components of the local SMB is considered a difficult task (King and Turner, 1997; Vaughan et al., 1999; van Lipzig et al., 2002; van den Broeke et al., 2004; Eisen et al., 2008; Gorodetskaya et al., 2015; Amory et al., 2017; Grazioli et al., 2017a).

Traditional snowfall measurements using precipitation gauges are inhibited by high wind speeds over the AIS leading to undercatchment, whereas ice cores and stake measurements are poor proxies for snowfall as they are affected by other components of the local SMB, such as e.g. blowing snow (see Eq. 1; Bromwich, 1988; van den Broeke et al., 2004; Groot Zwaaftink et al., 2013; Das et al., 2013). Again this indicates the importance of the separate measurement of the snowfall component and the existence of a non-linear relation between the local SMB and snowfall amounts. In recent years, remote sensing applications offer new possibilities regarding the determination of snowfall amounts on remote locations such as the AIS. A first estimate of snowfall rates over the AIS was derived from the Cloudsat satellite (Palerme et al., 2014). Its low overpass frequency, narrow swath width and ground clutter remains a limiting factor (Battaglia and Delanoë, 2013; Maahn et al., 2014; Casella et al., 2017). More locally, the operation of ground-based precipitation radars has proven to be an efficient way to detect snowfall over the AIS at several locations (Gorodetskaya et al., 2015; Souverijns et al., 2017; Grazioli et al., 2017a, b).

Wind-induced accumulation and ablation by blowing or drifting snow over the AIS has an important impact on the local SMB (Bromwich et al., 2004; Scarchilli et al., 2010; Palm et al., 2017). The $ER_{ds}$ component can however only be measured accurately using a network of snowdrift instrumentation (Nishimura and Nemoto, 2005; Leonard et al., 2012; Nishimura and

Ishimaru, 2012; Barral et al., 2014; Trouvilliez et al., 2014; Amory et al., 2017) and is difficult to take into account in Antarctic-wide SMB estimates (Genthon and Krinner, 2001; van den Broeke et al., 2004; Das et al., 2013). Atmospheric models have been adapted to take these processes into account (Lenaerts and van den Broeke, 2012a; Gallée et al., 2013; Amory et al., 2015), but depend on parametrisations. Thresholds for drifting snow initiation are very variable and depend on both the meteorological

conditions and the snow particles characteristics (Li and Pomeroy, 1997; Bintanja and Reijmer, 2001). Blowing and drifting snow can shape the relief of the ice sheet on the small spatial scale (i.e. sastrugi formation), local scale (i.e. barchan dunes) or the kilometer scale (i.e. blue ice zones) (Amory et al., 2017). These processes potentially attribute for a large variability in snow height on the decameter scale (Libois et al., 2014). Due to the difficult measurement of $ER_{ds}$, it is often considered as the leftover term in the local SMB over the AIS (Gorodetskaya et al., 2015). Recently, some new remote sensing techniques

have been developed to detect $ER_{ds}$ based on satellite-borne lidar and ground-based ceilometer measurements, adding to the understanding of the $ER_{ds}$ component in the SMB (Palm et al., 2011; Gossart et al., 2017).

Both surface and drifting snow sublimation have been quantified for the PE station (Thiery et al., 2012). At the local scale, the significance of the processes can be fairly large (e.g. King et al., 2001; Bliss et al., 2011; Gorodetskaya et al., 2015; Grazioli et al., 2017b). For the PE station, sublimation was found to remove 10 % of the total precipitation (Thiery et al., 2012). In

this study, the focus is mainly on the relation between accumulation, ablation and snowfall. The sublimation terms, together with melt which is only relevant at coastal areas and ice shelves (Lenaerts et al., 2017), are therefore only quantified and not investigated in great depth.

Snow height (changes) can be measured by an Automatic Weather Station (AWS), which is equipped with an acoustic height ranger (van den Broeke et al., 2004). The main advantage of the AWS is the combination of snow height and meteorological

observations.

Synoptic and meso-scale meteorology have a strong impact on the local SMB. For example, cyclone activity in the Antarctic circumpolar trough (50°-70° S) attributes for moist air penetrating into the atmosphere above the AIS. This leads to snowfall events and high wind speeds at both coastal and inland locations (King and Turner, 1997; Simmonds et al., 2003; Schlosser et al., 2010; Hirasawa et al., 2013; Gorodetskaya et al., 2013). In case large amounts of moisture are transported, these events

are identified as atmospheric rivers having a profound impact on the local SMB (Gorodetskaya et al., 2014). Nevertheless, the independent measurement of the snowfall component and the local SMB over the AIS are limited. Gorodetskaya et al. (2015) showed that snowfall events at the PE station do not necessarily contribute to accumulation or an increase in snow height.

This study uses 37 months of independent snowfall and SMB measurements in order to assess statistically the relation between snowfall and the local SMB. This analysis allows therefore to determine the representativeness of stake measurements

as a proxy for snowfall at the local scale and to define the frequency and conditions when snowfall leads to accumulation or ablation. In addition, changes in the local SMB take place without snowfall. A thorough analysis of meteorological conditions during these events adds to the understanding of the behaviour of the $ER_{ds}$ component and their impact on the evolution of the local SMB over the AIS.

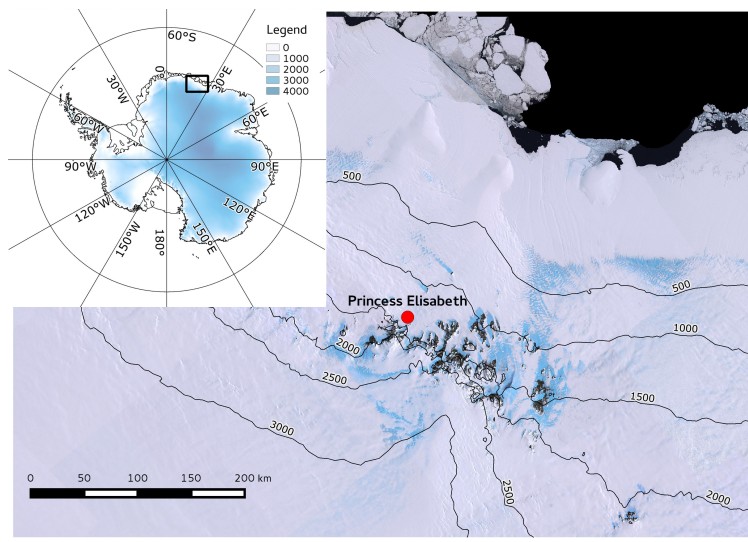

**Figure 1.** Location of the meteorological observatory at the Belgian Princess Elisabeth station, north of the Sør Rondane Mountains. Source: Landsat Image Mosaic of Antarctica (LIMA) Project, http://lima.usgs.gov/. Colors in the inset and contours on the main figure denote topography height (meters).

## 2  Material and methods

### 2.1  Site description and instrumentation

Long-term measurements of the individual components of the SMB over the AIS are scarce due to its harsh environment and difficult accessibility. To tackle this problem, a limited-maintenance and low-power meteorological observatory was established
at the PE station in 2009 (Gorodetskaya et al., 2015). The PE station is located in the escarpment zone of the East Antarctic plateau (71°57' S, 23°21' E; 1392 m above sea level), 173 km from the coast, in Dronning Maud Land, north of the Sør Rondane mountain chain (Fig. 1). Meteorological conditions at the station are influenced by both synoptic weather conditions and katabatic winds. A detailed description of the site can be found in Pattyn et al. (2010) and Gorodetskaya et al. (2013). The main aim of this meteorological observatory is to collect a comprehensive database of meteorology, radiative fluxes, snow
height changes, and cloud and precipitation properties (Gorodetskaya et al., 2015, http://www.aerocloud.be).

Snowfall measurements are recorded since 2010 by use of a vertically pointing Micro Rain Radar-2 (MRR) operating at a frequency of 24 GHz (Klugmann et al., 1996; Peters et al., 2002). The potential of millimetre radars to efficiently detect snowfall has been shown by Matrosov et al. (2008) and has been evaluated specifically for this type of radar by Kneifel et al. (2011). As the MRR was originally developed for rain observations, operational MRR procedures to derive standard radar
variables like effective reflectivity factor ($Z_e$) or mean Doppler velocity had to be modified for snowfall using the methodology of Maahn and Kollias (2012). In order to obtain reliable estimates of snowfall rates and their uncertainty, an optical disdrometer (Precipitation Imaging Package Newman et al., 2009) was installed at the PE station. This instrument is based on a high speed

camera and is able to obtain detailed information about snowflake microphysics (i.e. the particle size distribution). A correction for the horizontal and vertical displacement of snowfall between the MRR data acquisition level and the surface has been applied (Wood, 2011; Souverijns et al., 2017). Using this information, a relation between radar reflectivity measured by the MRR and snowfall rates was obtained: $Ze = 18SR^{1.1}$. Furthermore, a constraint on the uncertainty of the resulting snowfall rates

was obtained [-59 % +60 %] (10th-90th percentile) (Souverijns et al., 2017), which is a considerable reduction compared to earlier snowfall rate estimates at the PE station that were retrieved without any information on the snow particle microphysical characteristics (Gorodetskaya et al., 2015).

The net local SMB is calculated based on snow height measurements of an AWS and yearly snow density records (see Section 1). The AWS was installed in February 2009 and is located approximately 300 m east of the station (and the MRR)

on the windward side of Utsteinen ridge (see Section 2.1) in a zone of little accumulation (Pattyn et al., 2010). The AWS records meteorological variables, such as air temperature, pressure, wind speed and direction, relative humidity and radiative fluxes at 2 meters above the surface. These measurements are processed following Gorodetskaya et al. (2013). Wind speed and directions are recorded by an anemometer at the top of the AWS. Humidity is recorded with respect to water at the top of the AWS and is converted to humidity with respect to ice using the conversion of Anderson (1994). Broadband radiative

fluxes are measured using pyranometers and pyrgeometers. All the above parameters are measured with 6-minute resolution and averaged to hourly means. Furthermore, the AWS is equipped with an acoustic height sensor, which measures snow height changes once an hour with an accuracy of 1 cm on an hourly time resolution. A running mean of 24 hours is applied to erase the majority of the short-term decameter scale variability due to sastrugi movement (Libois et al., 2014). Nevertheless, few events may still be related to artefacts caused by the motion of sastrugis. As such, temporary peaks (i.e. with a strong decay within 48

hours) in the snow height records are excluded from the analysis. In January 2016, a new AWS was set up to replace the one installed in 2009, able to measure snow height changes more accurately. A detailed overview of the specifications of the old AWS including its uncertainty can be found in Gorodetskaya et al. (2013) and of the new AWS in the Supplement (Table S1). Each year, the average density of the snow that has accumulated in the past year is calculated from snow pit measurements at the PE station. Using these average yearly densities (varying between 309-375 kg m$^{-3}$), snow height changes are converted to

changes in the local SMB (water equivalent) (Gorodetskaya et al., 2013, 2015). Densification due to snow aging is not taken into account.

Ablation or accumulation attaining for changes in the SMB occur generally in blowing snow layers with a distinct vertical extent. Recently, an algorithm for the detection of blowing snow by the use of a ceilometer was developed (Gossart et al., 2017). Ceilometers are ground based lidars originally developed to detect cloud base height (Van Tricht et al., 2014; Gossart et al.,

2017). This instrument is able to detect blowing snow particles based on the backscatter signal and is operational during all types of weather conditions (cloudy, snowfall conditions and at night). As such, it complements satellite retrievals of blowing snow from the CALIPSO satellite (Palm et al., 2011). The minimal height of the blowing snow layer to be detected by the ceilometer equals 30 m at the PE station. The temporal resolution equals 15 sec, while the vertical height resolution is 10 m. The quantification of blowing snow particle density, shape or number from the ceilometer attenuated backscatter signal is very

uncertain (Wiegner et al., 2014). As such, this study is limited to the determination of the blowing snow frequency. Results

show that 78 % of the detected events by the ceilometer are in agreement with visual observations at Neumayer III station and that blowing snow occurrence strongly depends on fresh snow availability in addition to wind speed (Gossart et al., 2017).

Surface and snowdrift sublimation are quantified using the approach of Thiery et al. (2012), which determine their magnitude based on parametrisations using meteorological observations such as relative humidity and wind speed (e.g. Bintanja and Reijmer, 2001; Déry and Yau, 2001). The necessary data is provided by the AWS. Based on the AWS accuracy, the uncertainty of the surface sublimation term equals approximately 47 %, while for snowdrift sublimation, it equals 20 % based on differences between parametrisations (both two standard deviations) (Gorodetskaya et al., 2015).

The sampling period of our study is limited to time periods when both the AWS and MRR were operational between 2010-2016. These time frames are almost always restricted to austral summer months when the station is manned (Fig. S1 in the Supplement). Data acquisition during the austral winter season is often hampered by power failures at the PE station. The MRR was installed in the season 2009-2010 and a total record of seven austral summer seasons is available for analysis. For austral winter, only one season is available where both MRR and AWS were operating continuously i.e. 2012. Apart from this, in January 2015, a problem with the wind vane of the AWS was detected. This problem persisted until December 2015, leading to erroneous wind direction measurements. The year 2015 is therefore also discarded from our analysis. In the end, 37 months of collocated precipitation, total SMB and meteorological data are available at the same location, which is unprecedented for the Antarctic region.

## 2.2 Snowfall, accumulation and ablation events

The local SMB is composed of the sum of different components (Eq. 1). Snowfall amounts are measured by a Micro Rain Radar, while the two sublimation terms are calculated based on meteorological data from the AWS (see Section 2.1). The $ER_{ds}$ term is not measured directly at the PE station. Nevertheless, since the total SMB can be deduced from the AWS (see Section 1), $ER_{ds}$ can be calculated as the residual term after inverting Eq. 1 (Gorodetskaya et al., 2015). Its uncertainty is therefore based on the uncertainty of all other components of the local SMB (Eq. 1; Gorodetskaya et al., 2015) and is mainly determined by the uncertainty of the snowfall component at the PE station, as this term has the highest uncertainty range (see Section 2.1) and absolute contribution to the SMB.

Snowfall is generally considered the main positive term of the local SMB (Davis et al., 2005). One would therefore expect that snowfall results in accumulation or an increase in height. However, this is not necessarily true locally over the AIS, where snowfall often coincides with strong winds. Consequently, snowfall events with strong redistribution can result in a net snow removal at the local scale. As stated in section 1, accumulation and ablation also occur during non-precipitating conditions. In order to define the conditions for these episodes and to assess a relation between snowfall and accumulation, different types of events were discriminated based on local SMB measurements from the AWS and snowfall records from the MRR:

- Accumulation during snowfall (SMB +, S +)

- Ablation during snowfall (SMB -, S +)

- Accumulation without snowfall (SMB +, S 0)

– Ablation without snowfall (SMB -, S 0)

An event is confined to the period between the start of snowfall, accumulation or ablation and the moment no more snowfall, accumulation or ablation is observed. The time step and duration of an event have an hourly time resolution. Snowfall events are defined as exceeding the threshold of 1 mm w.e. during the continuous duration of snowfall measured by the MRR. As the AWS measures snow height changes with a sensitivity of 1 cm, an increase in surface height larger than 1 cm over the duration of an event is considered to be an accumulation event, while an ablation event occurs when a decrease in surface height of more than 1 cm is recorded.

## 2.3  Local and large scale meteorology

In order to understand the mechanisms attaining for snowfall and wind-induced accumulation or ablation, the meteorological conditions of the four types of events defined in Sect. 2.2 are evaluated. For all members within these four types of events, average meteorological conditions are calculated. Wind speed and direction, humidity and radiative fluxes are obtained from the AWS. Snowfall amounts are obtained by applying a relation converting radar reflectivity measurements from the MRR to snowfall rates specifically developed for the PE station using observations from an optical disdrometer (see Section 2.1).

Further, the temporal extent of the cloud system is investigated using two different data products. Firstly, cloudy conditions are estimated based on longwave downward radiation measurements of the AWS. For this, the Clear Sky Index based on the methodology of Marty and Philipona (2000) and Dürr and Philipona (2004) is used, which calculates the ratio between the apparent and clear-sky longwave downward radiation. The apparent longwave downward radiation is calculated using observations of the AWS. For the clear-sky longwave downward radiation, local coefficients (see Eq. 3 and Sect. 3 in Marty and Philipona (2000)) are optimised for the PE station by comparing calculated clear sky and observed longwave downward radiation during visually detected cloud free conditions (based on camera images and ceilometer data). As a result, based on the ratio of apparent and clear-sky longwave downward radiation, one can discriminate between cloudy and clear-sky conditions. Secondly, the temporal extent of the cloud system at the PE station was estimated from ERA-Interim (Dee et al., 2011). ERA-Interim is generally considered the best reanalysis product over Antarctica (Bracegirdle and Marshall, 2012). Cloudy conditions over the station are defined as having a total cloud fraction of more than 95 % in the pixel over the PE station. Sensitivity studies have been executed on this threshold (varying between 80 and 100 %) and account for only small relative differences not influencing the general conclusions significantly. Ceilometer cloud observations are not used to determine the temporal extent of the cloud system, as the collocated time period of corresponding measurements is insufficient.

Apart from these local meteorological variables, an analysis of the large-scale circulation over Dronning Maud Land (including a large part of the Southern Ocean) was performed. A cluster analysis was applied to 500 hPa geopotential ERA-Interim reanalysis data covering the period of observations at the PE station (2010-2016). This gives an overview of the climatology and the typical circulation that is present over the region.

Several algorithms are available to perform a cluster analysis (Philipp et al., 2010). In recent studies, thorough evaluations were performed for each of these algorithms, indicating the best performance regarding circulation clustering for optimisation

algorithms over different parts of the world (Huth et al., 2008; Beck and Philipp, 2010; Casado et al., 2010; Souverijns et al., 2016). From these optimisation algorithms, the simulated annealing and diversified randomisation (SANDRA) algorithm was chosen, which is based on k-means clustering (Philipp et al., 2007) as it performed adequately for different applications over the world regarding circulation clustering (Huth et al., 2008; Beck and Philipp, 2010; Casado et al., 2010; Souverijns et al., 2016). Next to the choice of the classification algorithm, it is also necessary to define the total number of circulation patterns that covers the full climatology over Dronning Maud Land. For a range between 2 and 27 total circulation patterns, the quality of the SANDRA algorithm is tested. Using the Fast Silhouette Index, the ability of the SANDRA algorithm to maximise the separability between the members of different circulation patterns, while minimising the variances within each circulation pattern, was investigated (Rousseeuw, 1987). In this study, a total of six circulation patterns was selected. The Fast Silhouette index indicates a local minimum value as a further increase in the total number of circulation patterns shows no significant improvement (i.e. decrease) regarding the classification skill (Fig. S2). Based on this climatology, the dominant circulation present during the four types of events attaining for a change in the local SMB defined in Sect. 2.2 are determined. This cluster analysis method gives similar results to the local meteorological regimes defined by Gorodetskaya et al. (2013), but increases insight into the large-scale circulation patterns and source regions of air advection.

Next to this cluster analysis, a backtrajectory analysis was performed for all individual events in order to get insights into the origin and history of the air masses. For this, the FLEXPART software, a Lagrangian transport and dispersion model which makes use of ERA-Interim data, is used at a spatial resolution of $0.75° \times 0.75°$ (Stohl et al., 1995; Stohl and Seibert, 1998).

Changes in snow height are measured by the AWS at a single location, 300 m east of the PE station. However, snow height is highly variable in space and time over the AIS (Frezzotti et al., 2005; Eisen et al., 2008). Observations from the Antarctic plateau show large decameter-scale variability in snow heights due to the formation and movement of sastrugi, highly impacting the snow height record (Libois et al., 2014). In this study, the focus is mainly on accumulation and ablation over larger spatial areas. In order to validate the spatial scale of accumulation and ablation, snow erosion output from the state-of-the-art high resolution (5.5 km grid) RACMO2.3 simulation is analysed (Lenaerts et al., 2014; van Wessem et al., 2014; Lenaerts et al., 2017), which is coupled to a snow model including drifting snow (Lenaerts and van den Broeke, 2012a). The RACMO2.3 simulation is driven by ERA-Interim and adequately simulates both climatological and meteorological conditions near the surface (e.g. Lenaerts et al., 2014). By selecting the time periods during which accumulation and ablation occurred in the RACMO2.3 simulation, an overview of snow erosion spatial structures during these event is achieved.

## 3 Results and discussion

### 3.1 Local surface mass balance

The four components of the local SMB, snowfall, surface sublimation, drifting snow sublimation and wind-induced accumulation and ablation, are converted to water equivalent values (see Sect. 1; Gorodetskaya et al., 2013, 2015). When snow height measurements are available the local SMB can be closed (treating $ER_{ds}$ as a residual term; see Eq. 1). For the year 2012, both the AWS and MRR were operating year-round, allowing to visualise the evolution of the different components through time in

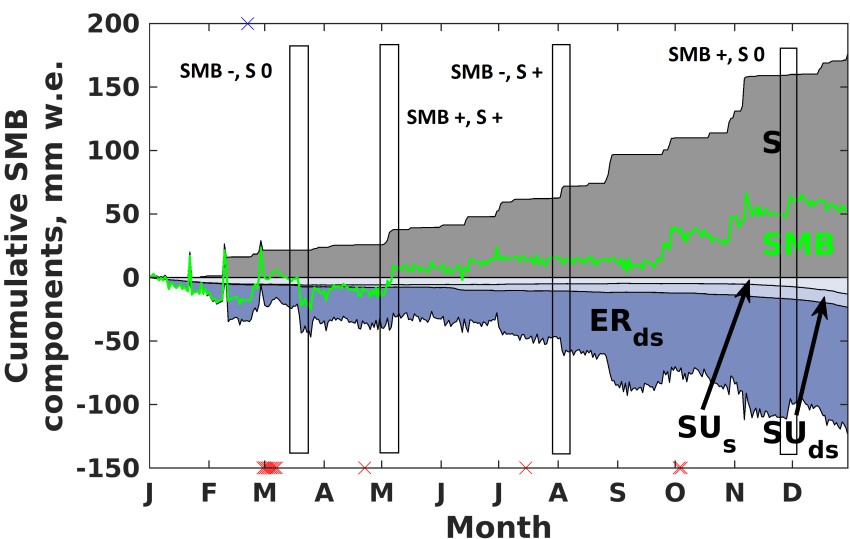

**Figure 2.** Cumulative daily surface mass balance components during 2012 at the Princess Elisabeth station: snowfall (S), surface sublimation ($SU_s$), drifting snow sublimation ($SU_{ds}$), wind-induced accumulation and ablation ($ER_{ds}$), and accumulation and ablation deduced from measured snow height changes since 1 January 2012 (SMB; adapted from Gorodetskaya et al. (2015)). $SU_s$ and $SU_{ds}$ are plotted as ablation terms. $ER_{ds}$ is calculated and plotted as a residual term by inverting Eq. 1. Red crosses at the bottom indicate days of missing MRR data, while blue crosses at the top denote missing AWS data. Letters on x axis mark the first day of each month. Examples of the four types of events defined in Sect. 2.2 are indicated by black rectangles.

a cumulative way (Fig. 2; see also Gorodetskaya et al. (2015)). Snowfall (S) is identified as a strictly positive term and the step-wise function indicates that the total yearly precipitation amount is strongly influenced by several intense precipitation events per year. The local SMB is denoted by the green line and shows several peaks in both the upper and lower direction (indicating distinct accumulation and ablation events respectively). These events occur both with or without snowfall and allows for easy

5    identification of the individual accumulation and ablation events. Furthermore, it can also be noted that surface and drifting snow sublimation ($SU_s$ and $SU_{ds}$ respectively) are both ablation terms. The rest of the study ignores these two components, as in this study we focus on the contribution of the $ER_{ds}$ term.

For each of the four types of events defined in Sect. 2.2 one example is highlighted in Fig. 2. It can be seen that some snowfall events account for accumulation (SMB +, S +), but that this is not strictly the case (Gorodetskaya et al., 2015). Wind can also

10    remove mass from the site during a snowfall event, leading to a net removal of mass (SMB -, S +). Additionally, even during time periods without any snowfall, snow height varies continuously and several accumulation (SMB +, S 0) and ablation (SMB -, S 0) events can be identified without the presence of snowfall.

## 3.2 Large-scale meteorology

Using the SANDRA circulation pattern classification algorithm, it is possible to define the climatology of large-scale circulation over Dronning Maud Land and the nearby Southern Ocean (Fig. 3). Large-scale circulation over Dronning Maud Land is typically characterised by an anticyclone close to the pole and cyclones at latitudes between 50° S and 70° S, north or near the coast of the AIS. These low pressure systems form a ring around the Antarctic continent (Antarctic circumpolar trough), attaining for a high variability in meteorological regimes at the coastal areas of the AIS (King and Turner, 1997; van den Broeke and van Lipzig, 2003; König-Langlo and Loose, 2006; Gorodetskaya et al., 2013; Hirasawa et al., 2013). The sequence of circulation patterns depicted in Fig. 3 is typical for the Antarctic region (Simmonds et al., 2003). In circulation type 1 (C1) the Antarctic circumpolar trough is clearly visible showing a high pressure bulge over Dronning Maud Land. East and west, we can see two low pressure cells. In C2-C5 the typical movement of a low pressure cyclone from the west to the east is depicted, largely influencing meteorological conditions at the surface. Apart from the circulation, also the average precipitation amounts associated to each circulation pattern are shown (Fig. 3). Precipitation estimates are obtained from ERA-Interim, currently considered the best Antarctic-wide precipitation product, however still strongly biased (Bromwich et al., 2011). Misrepresentations of large-scale atmospheric flow and precipitation might therefore impact the results. A strong link between the location of the cyclone and precipitation amounts is present. The cyclone is capable of transporting marine air towards the AIS. These marine air masses have the potential to take up moisture, potentially attaining for precipitation at the continent. A detailed description of each circulation pattern individually can be retrieved from the Supplement.

## 3.3 Snowfall

Snowfall is the main positive contributor to the local SMB. During the observational period 2010-2016, in total 50 independent snowfall episodes were detected attaining for at least 1 mm w.e. at the PE station. Snowfall events are characterised by high wind speeds originating mainly from the East North East (ENE; Table 1 & Fig. 4a). Only a limited number of snowfall events correspond to near-surface winds from other directions. This complies with literature, stating that moist air and precipitation is transported via cyclone activity in the Antarctic circumpolar trough towards the AIS (Sect. 1) and is also confirmed by our cluster analysis. A fraction of more than 70 % of all snowfall events coincide with cyclone activity (Fig. 4b). Over the AIS, these winds are slanted towards the east at the surface due to friction and the location of the PE station in the escarpment zone (Fig. 1).

In order to validate the relation between large snowfall events and these meteorological conditions, interactions between the transport capacity of the cyclone, the origin of the air mass and the amount of snowfall that is recorded at the PE station are investigated. The transport capacity of the cyclone is parametrised by an index we constructed based on the pressure difference between the PE station and the typical location of the trough northwest of the station (0° E, 62° S, Fig. 3) as these cyclones attribute for the highest snowfall amounts at the PE station. Larger values for this index indicate higher pressure gradients and stronger wind speeds. The origin of the air masses (five days prior to the event) are deduced from backtrajectories arriving at the PE station at altitudes below 3000 meters a.s.l.. The most intense snowfall events (> 5 mm w.e. of snowfall per event) are

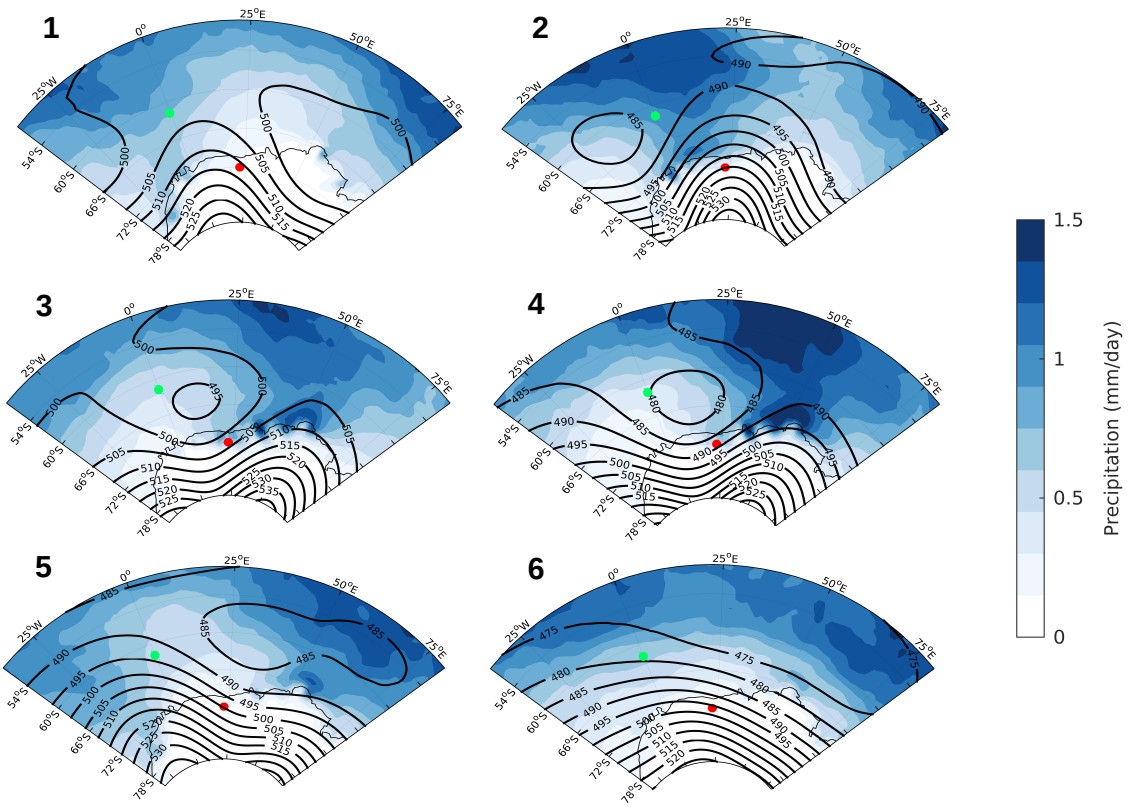

**Figure 3.** Weather atlas illustrating the circulation climatology over Dronning Maud Land. Thick lines denote the 500 hPa geopotential fields, while blue colours show average precipitation amounts linked to this circulation. The red dot indicates the location of the Princess Elisabeth station, while the green dot denotes the location over the ocean used for calculating the pressure gradient (Section 3.3). The Antarctic circumpolar trough is identified. The depiction sequence of the circulation types describes the typical west-east movement of the cyclones.

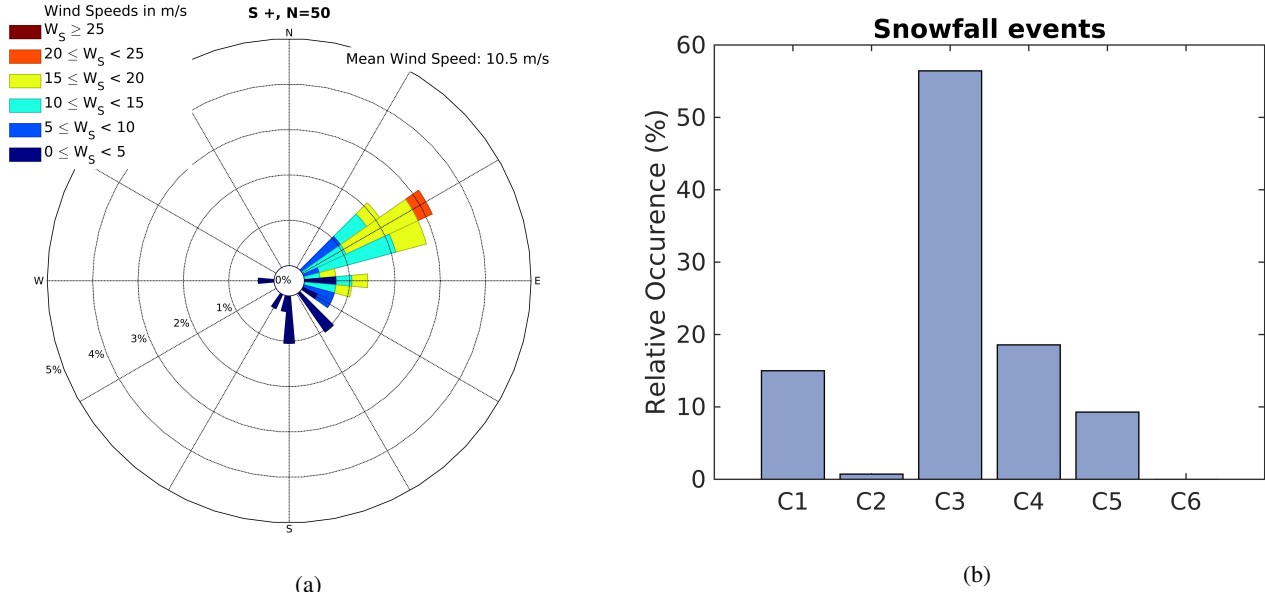

**Figure 4.** (a) Wind rose showing the hourly speed and direction of the wind recorded by the AWS during snowfall events detected by the MRR. N denotes the total number of snowfall events during the observation period. (b) Frequency of occurrence of the circulation patterns depicted in Fig. 3 for snowfall events detected by the MRR.

| | SMB +, S + | SMB -, S + | SMB +, S 0 | SMB -, S 0 |
|---|---|---|---|---|
| Wind Speed (ms$^{-1}$) | 10.6 | 10.5 | 6.70 | 5.44 |
| Wind Direction (° deviation from N) | 87.7 | 83.8 | 108 | 146 |
| Station Pressure (hPa) | 830 | 830 | 828 | 829 |
| Relative Humidity (%) | 91.3 | 89.5 | 67.2 | 60.1 |
| Snowfall summed over the event (mm w.e.) | 5.25 | 3.56 | - | - |
| Deviation from LW$_{cs}$ (W m$^{-2}$) | 77.0 | 72.3 | 36.0 | 25.0 |
| Deviation from SW$_{cs}$ (W m$^{-2}$) | 109 | 83.2 | 71.1 | 34.0 |
| Temporal extent of cloud system ERA-Int (hours) | 46.8 | 32.4 | 31.8 | 27.0 |
| Temporal extent of cloud system AWS (hours) | 42.6 | 29.9 | 32.6 | 27.6 |
| Total duration of the event (hours) | 16.9 | 12.3 | 12.3 | 11.6 |
| Number of events | 31 | 19 | 72 | 87 |

**Table 1.** Average meteorological statistics for the four types of events stated in Sect. 2.2. Only events during coinciding measurements of the AWS and MRR are included. LW$_{cs}$ denotes the longwave incoming radiation during clear-sky conditions, while SW$_{cs}$ is the shortwave incoming radiation during clear-sky conditions.

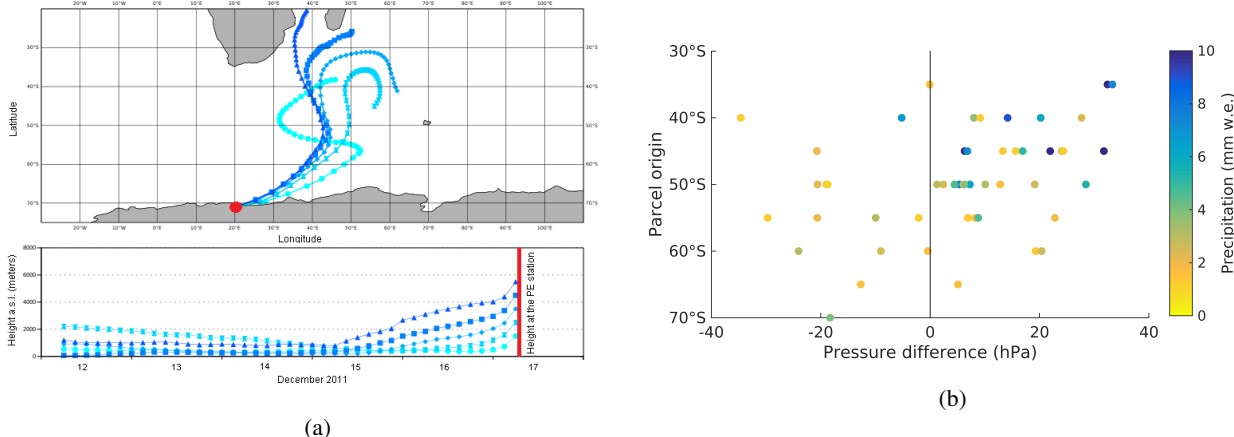

(a)

(b)

**Figure 5.** (a) Backtrajectories for the snowfall event of the 17th of December 2011 at the Princess Elisabeth station for different heights (start at 06 UTC). The red dot (top panel) denotes the location of the Princess Elisabeth station, while the red line (bottom panel) shows the time the snowfall event arrived at the station. (b) Relation between the transport capacity of the cyclones (based on the pressure difference between the Princess Elisabeth station and $0°$ E, $62°$ S), the origin of the air masses (based on the origin of the backtrajectories arriving at the Princess Elisabeth station at altitudes below 3000 meters a.s.l. five days prior to the event) and the total amount of snowfall for all detected events at the Princess Elisabeth station during 2010-2016 . Positive values for the pressure difference denote the presence of a cyclone northwest of the Princess Elisabeth station, while negative values indicate the absence of a cyclone.

associated with air masses typically originating from areas north of $50°$ N, taking up moisture close to the oceanic surface and are generally lifted upwards when reaching the AIS continental margin (An example is given in Fig. 5a). A significant relation between the transport capacity of the cyclone, the origin of the air mass and the amount of snowfall is observed (Fig. 5b). In case no cyclone is present NW of the PE station (i.e. negative pressure difference values; left side of the graph), snowfall

amounts are generally low. This corresponds to events that are not related to winds originating from the ENE (Fig. 4a) and confirms that snowfall amounts that are not related directly to cyclones are low. In case a cyclone is present, a tendency for higher snowfall amounts during larger pressure gradients is present (significant at the 0.05 confidence level). Furthermore, during these conditions, air masses originate from more northern areas (Fig. 5b). As a conclusion, when the cyclone or trough is more developed and high pressure blocking is present NE of the PE station, moisture from more northern areas is able to

be transported, leading to higher snowfall rates at the station. In case very high snowfall amounts are observed during this synoptic situation, moisture transport is likely related to atmospheric rivers (Gorodetskaya et al., 2014).

From the total number of snowfall events, 31 resulted in accumulation (62 %, attaining on average for 5.3 mm w.e. of snowfall), while 19 led to ablation (38 %, attaining on average for 3.6 mm w.e. of snowfall; Table 1). Regarding the basic meteorological variables obtained by the AWS, no large differences between both types of events are observed. Note especially

that the mean wind speed is the same. Only the duration of the event and the temporal extent of the cloud structure show a clear distinction regarding their mean values. Accumulation events (SMB +, S +) have a longer duration compared to ablation events (SMB -, S +) significant at the 0.1 level (Fig. 6). Furthermore, the persistence of the cloud structure shows a notable

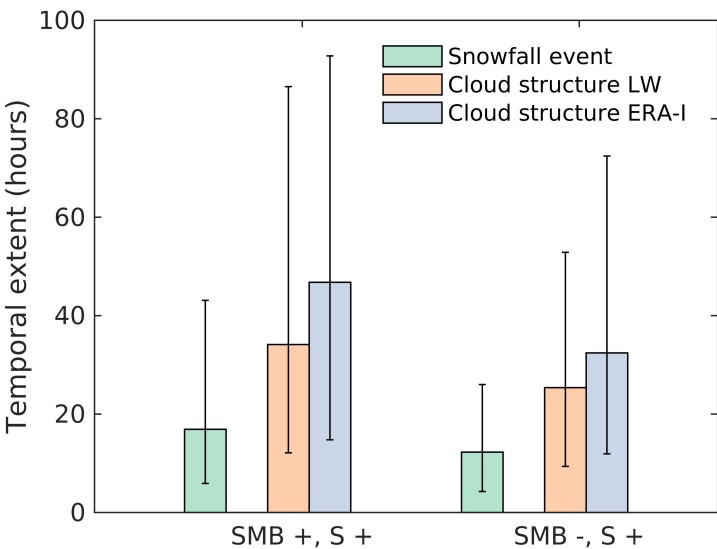

**Figure 6.** Duration (green) and the temporal extent of the cloud structure (red and blue) for snowfall events attaining for accumulation (SMB +, S +) and ablation (SMB -, S +) (hours). Columns denote the average value, while the error bars denote the 25th and 75th percentile based on all events.

difference for both methods described in Sect. 2.3 at the 0.05 level (Table 1 & Fig. 6). It is noted that relatively speaking, the temporal extent of the cloud structure equals on average 260 % of the duration of snowfall for both the accumulation and ablation events using both methods. It is therefore concluded that accumulation events are characterised by a larger temporal extent of the event compared to ablation, which is deduced from both the duration of snowfall and the temporal extent of the

5 cloud structure.

Snowfall events are generally characterised by high wind speeds originating from oceanic areas (ENE at the PE station) redistributing snow continuously during the time of an event, including a mix between blowing snow and precipitation particles (Nishimura and Nemoto, 2005). In case a snow storm has a large temporal extent, fresh snow precipitates over areas with a large spatial coverage. Therefore, there is a higher chance that snow removed from the PE station is replaced by fresh snow

from other areas located upstream. In case of a shorter event or cloud system, there is a lower probability that snow from other locations replaces snow that is blown away from the station. As such, the larger the cloud system or the duration of the event, the higher the chance that snowfall events result in accumulation at a local scale, rather than ablation. In order to validate this hypothesis, information about the scale of snow displacement is necessary (see Section 3.4).

Redistribution of snow at the surface by high winds is denoted as blowing snow (drifting snow in case the height of the snow

displacement layer is lower than 2 meters). By using information from the ceilometer at the PE station, it is possible to detect blowing snow layers with a minimum vertical extent of 30 m (Gossart et al., 2017). The concurrence of blowing snow and snowfall events is analysed. For snowfall events attaining for accumulation, a blowing snow layer is detected in 95 % of the

cases, while for snowfall events attributing for ablation, this concurrence equals 88 %. In total, all 50 snowfall events attained for 542 cm of height changes (both accumulation and ablation; detected by the AWS). During the periods when the ceilometer recorded blowing snow, the AWS detected 486 cm of height changes (i.e. the sum of all height changes during events for which blowing snow was detected), showing the potential of the ceilometer to detect blowing snow during snowfall events.

During all snowfall events, a total amount of 230 mm w.e. was recorded. Based on yearly density measurements in the uppermost layers of the snow pack, yearly average values between 309-395 kg m$^{-3}$ are reported. This results in 58-75 cm accumulation at the surface if no other processes are at play. This is significantly lower than the total height changes recorded by the AWS during all snowfall events (542 cm). This indicates the importance of the continuous movement of snow during snowfall events. (Punctual) accumulation records are therefore not advised to be used as a proxy for snowfall over East Antarc-

tica. A direct relation between snowfall and accumulation is however difficult to assess, except that snowfall events of a longer duration have a higher chance to attain accumulation.

### 3.4 Wind-induced accumulation and ablation

In the previous section, synoptic events with snowfall lead to both accumulation and ablation, depending on the duration of the event (i.e. the duration of snowfall and temporal extent of the cloud structure). However, snow displacement events also occur

without the presence of snowfall. The meteorological conditions during these non-precipitating events are clearly different from snowfall events (Table 1), with clear differences in wind patterns (compare Fig. 7 & 4a). Firstly, the mean wind speed is much lower for non-precipitating events compared to snowfall events, while secondly, two dominant wind directions are detected. Thirdly, in contrast with snowfall events, clear differences between the wind patterns of accumulation and ablation events are visible (Fig. 7). On the one hand there is a dominant ENE flow during accumulation events (SMB +, S 0), while

southerly flow is more commonly related to ablation events (SMB -, S 0).

During accumulation events without snowfall (SMB +, S 0), one of the dominant wind directions is, just as during snowfall events, from the ENE. However, wind speeds are much lower (Fig. 7a). Approximately 40 % of all accumulation events occur during a similar circulation than snowfall events (circulation pattern 3 & 4 in Fig. 8a).

To assess the large-scale spatial patterns of snowdrift transport, our observed record is compared with output from a regional

climate model. Here, we use the high-resolution RACMO2.3 climate model output. In this simulation, a number of 36 modelled accumulation events attributed for a net snow accumulation of more than 1 kg m$^{-2}$ at the pixel corresponding to the PE station location in the RACMO2.3 simulation (which equals to height changes between 0.26-0.33 cm assuming snow densities between 309-375 kg m$^{-3}$), while 116 modelled ablation events attained for net snowfall removal of more than 1 kg m$^{-2}$. This is approximately 35 % of the observed quantities by the AWS. There are several potential underlying causes for this difference:

either, part of the snow erosion that is detected by the AWS occurs on the subgrid scale, or RACMO2.3 underestimates the blowing snow amount or its divergence. In fact, Lenaerts et al. (2014) showed RACMO2.3 underestimates blowing snow amounts around the PE station. Generally, RCMs tend to underestimate blowing snow amounts over the whole of Antarctica (Amory et al., 2015).

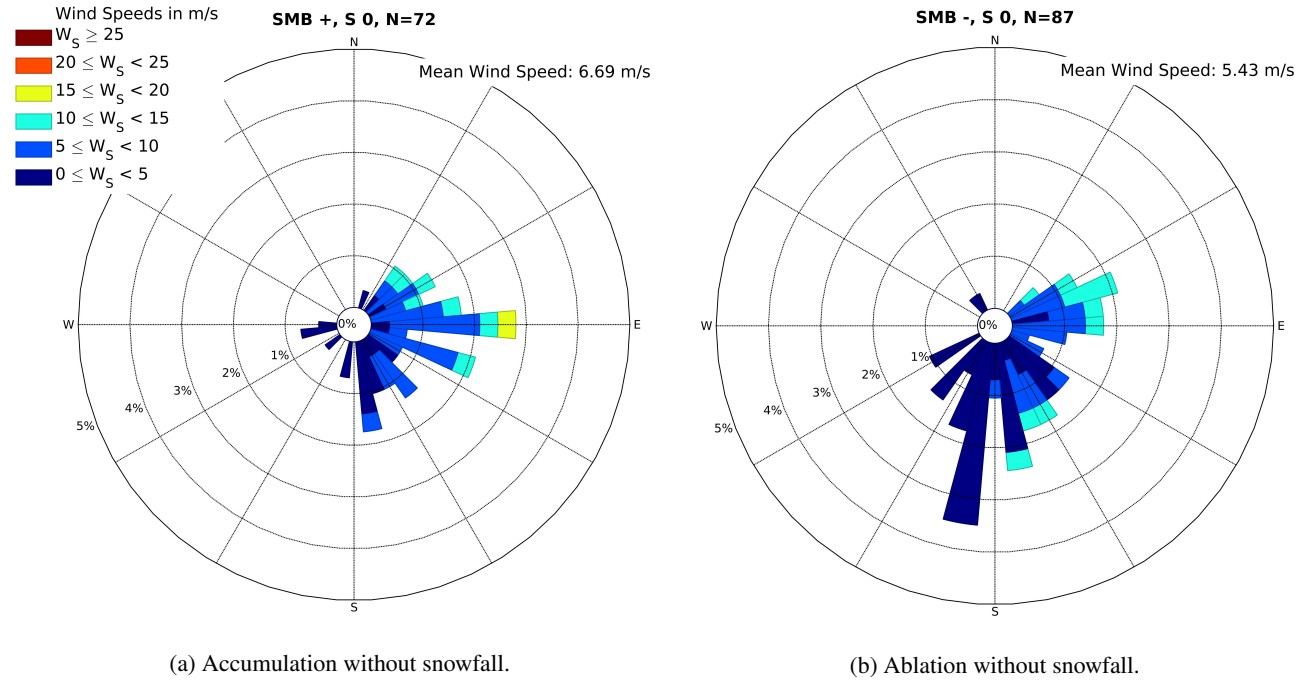

(a) Accumulation without snowfall.

(b) Ablation without snowfall.

**Figure 7.** Wind speed and direction for the two events attaining for wind-induced accumulation and ablation without snowfall. N denotes the total number of events during our observation period.

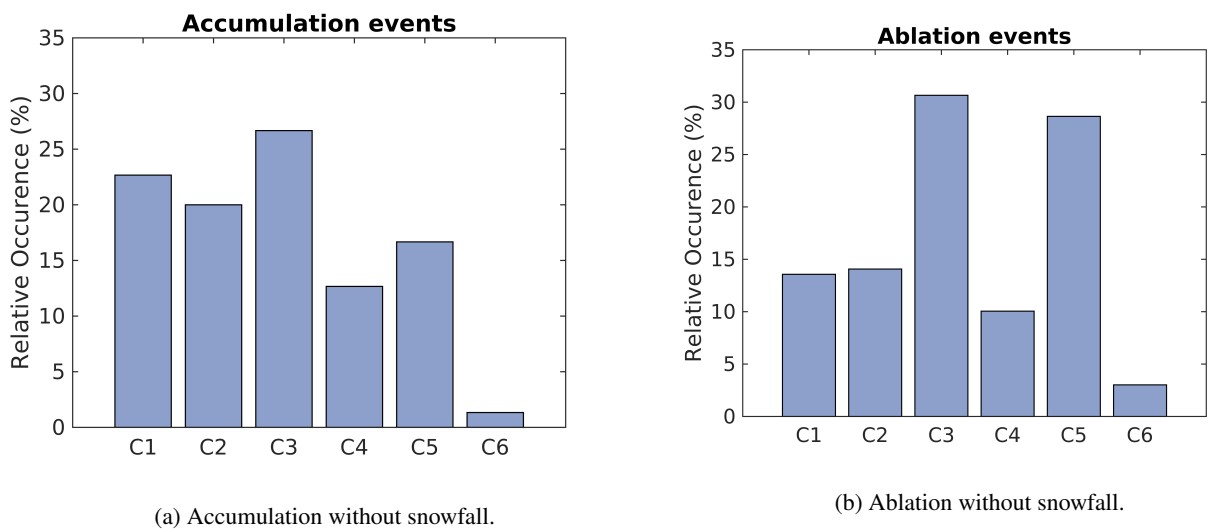

(a) Accumulation without snowfall.

(b) Ablation without snowfall.

**Figure 8.** Frequency of occurrence of the circulation patterns depicted in Fig. 3 for wind-induced accumulation and ablation without snowfall.

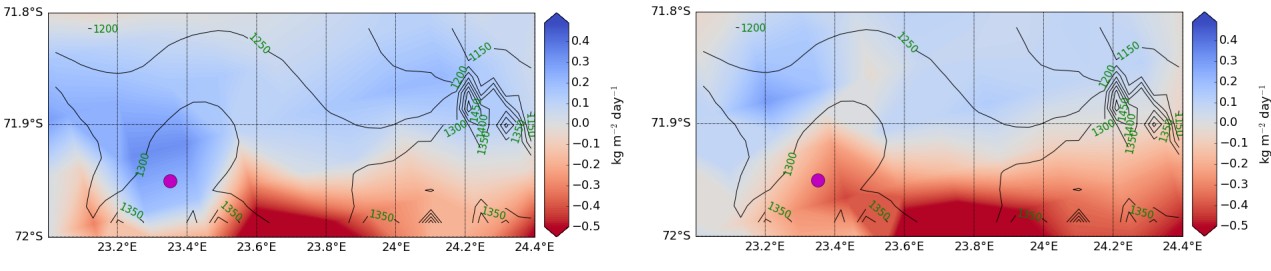

(a) Snow erosion during SMB +, S 0 at the PE station.

(b) Snow erosion during SMB -, S 0 at the PE station.

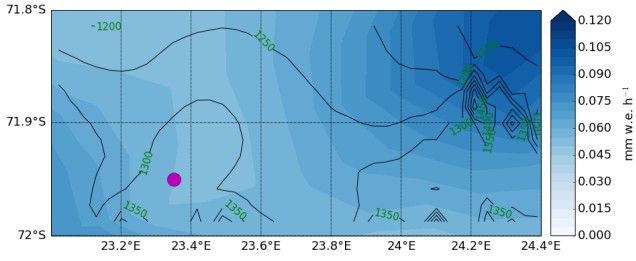

(c) Snowfall amounts during the 24 hours preceding the SMB +, S
and SMB -, S 0 events

**Figure 9.** Wind-driven snow accumulation and ablation during the two regimes at PE: a) accumulation regime without snowfall (SMB +, S 0) and b) ablation regime without snowfall (SMB -, S 0); c) average amount of snowfall during the 24 hours preceding both regimes. The fields are based on RACMO2.3 simulation. The purple dot indicates the location of the PE station, while the contours denote the topography in meters.

Almost all areas north of the mountain ridge generate snow accumulation in the RACMO2.3 simulation when there is accumulation but no snowfall at the PE station (Fig. 9a). A fraction of 43 % of accumulation events occurs within 24 hours after a snowfall event at the PE station (i.e. 31 out of 72 events). As shown by the RACMO2.3 simulation and ERA-Interim (compare Fig. 9c with Fig. 3 & 4b), most snowfall occurs upstream of the PE station, north of the Sør Rondane mountain ridge and ENE of the station during the 24 hours preceding the accumulation event at the PE station. This freshly fallen snow has a low density and is easily redistributed as the wind speed threshold for drifting snow erosion is low (Li and Pomeroy, 1997; Mahesh et al., 2003; Trouvilliez et al., 2014). As such, in case the wind pattern stays stable from the ENE after the snowfall, it can be transported upstream attaining for accumulation at more westerly locations, such as the PE station.

In case of ablation events (SMB -, S 0), the dominant wind direction originates from the south and is characterised by low wind speeds (Fig. 7b). The large-scale circulation analysis shows that there is a dominant occurrence of circulation pattern 5, apart from circulation pattern 3 (Fig. 8b). Circulation pattern 5 has only a limited influence of low-pressure systems. In order to understand these ablation events, their timing is investigated. A fraction of 46 % of the ablation events take place within 24 hours after a snowfall event. As stated in Sect. 3.3, snowfall events are mainly characterised by a ENE flow (Fig. 4a). After these snowfall episodes, the cyclone passes towards the east and winds tend to settle down (this evolution is depicted in Fig.

3). During calm conditions, a katabatic flow manifests originating from the mountain ridge and the Antarctic plateau (Parish and Cassano, 2003). These areas south of the PE station received generally less snowfall during the preceding snowfall event compared to the areas north of the Sør Rondane mountains (Fig. 9c & Palerme et al. (2014)). They are therefore capable of picking up fresh snow at the PE station attaining for ablation (Scarchilli et al., 2010). Similar conclusions can be drawn from

the analysis of the RACMO2.3 climate model output, showing large areas of snow removal at the edge of the mountain ridge (Fig. 9b) and was also identified in RACMO2.3 by Lenaerts and van den Broeke (2012b).

An analysis of the wind roses indicates the occurrence of accumulation and ablation at very low wind speeds (Fig. 7). This points out that the time since the last snowfall event and the availability of low-density fresh snow is of much higher importance than the wind speed in order to generate blowing snow, confirming the results of Gallee et al. (2001); Mahesh et al. (2003);

Scarchilli et al. (2010); Amory et al. (2017); Gossart et al. (2017). The time since the last snowfall event might also be an important parameter to explain the large variability in drifting snow wind speed thresholds and threshold friction velocities (Li and Pomeroy, 1997; Trouvilliez et al., 2014).

Based on the ceilometer, placed on a small ridge on the roof of the station, and its blowing snow detection algorithm, the concurrence of blowing snow and accumulation or ablation during non-snowfall conditions is analysed. For these accumulation

and ablation events, blowing snow is detected by the ceilometer in only 27 % and 20 % of the cases respectively. All changes in snow height due to accumulation and ablation attained for 1125 cm measured by the AWS. When the ceilometer detected blowing snow, only 274 cm of height changes were detected by the AWS, indicating that the blowing snow layer only has a limited vertical extent and that the transport of snow is restricted to shallow layers close to the ground during these type of events. Furthermore, it is noted that ablation and accumulation events can significantly compensate each other (the total

accumulation on a yearly basis due to snowfall and snowdrift is a lot smaller and has a large interannual variability ranging between 23 and 230 mm w.e. at the PE station).

Almost half of the accumulation and ablation events without snowfall occurred shortly after a snowfall event. Furthermore, wind speeds were found to be lower during these events compared to snowfall episodes. In both the accumulation and ablation case, fresh snow is easily transported towards or away from the station. During these types of events, the vertical extent of the

blowing snow layer does not generally reach 30 m and most of the transport of this freshly fallen snow including accumulation and ablation occurs in shallow layers at moderate wind speeds. As such, in order to get a good idea of accumulation and ablation during non-precipitating time periods and its influence on the local SMB, near-ground observations of drifting and blowing snow are indispensable (Takahashi, 1985; Nishimura and Nemoto, 2005; Bellot et al., 2011; Leonard and Maksym, 2011; Leonard et al., 2012; Nishimura and Ishimaru, 2012; Barral et al., 2014; Libois et al., 2014; Trouvilliez et al., 2014).

**4   Conclusions**

In this study, snowfall and associated surface mass balance (SMB) changes (accumulation and ablation) are investigated with regard to large-scale atmospheric circulation patterns at the Princess Elisabeth (PE) station in East Antarctica. Using a unique

set of remote sensing instruments, such as a Micro Rain Radar and an Automatic Weather Station, which operated concurrent for a period of 37 months, statistical relationships between meteorology and snow erosion are obtained.

Snowfall is the most important source term of the local SMB and originate from oceanic air masses which are transported towards the AIS by cyclones in the Antarctic circumpolar trough. A backtrajectory analysis showed that air masses originating
from more northern areas take up higher amounts of moisture which mainly precipitate at the coastal areas of the AIS. Because of high wind speeds associated with these events, displacement of freshly fallen snow takes place in layers with a vertical extent of usually more than 30 m as detected by the ceilometer at the PE station. The distinction between accumulation and ablation events during snowfall was correlated to the duration of the event (i.e. the duration of snowfall and the temporal cloud extent). Longer and larger events result in bigger areas with fresh snow deposition, allowing snow from synoptically upstream
areas to be transported towards the PE stations.

Wind-driven accumulation and ablation also occur without snowfall at the PE station. These accumulation events have a tendency to take place during similar types of circulation as snowfall events, however, they are characterised by lower wind speeds. During most of these accumulation events, snowfall took place upstream of the PE station, ENE of the station and to the north of the Sør Rondane mountain ridge, within the last 24 hours. Winds easily pick up the freshly fallen snow from upstream
areas and are capable of transporting it over tens of kilometers, leading to accumulation over large spatial areas potentially to higher elevations. Ablation events originate more often from southerly flows and also occur shortly after snowfall events at the station. Katabatic flow originating from the Antarctic plateau is dry, as it contains less snow particles, and removes the freshly fallen snow at the local site resulting in ablation. Both these accumulation and ablation events take place at low wind speeds in blowing and drifting snow layers of limited vertical extent as detected by the ceilometer at the PE station.

The results presented in this study may hint to transport of snow towards more inland locations, instead of the traditional view of transport towards the coast by strong katabatic winds. This mechanism is also seen in RACMO2.3 although the absolute magnitude of accumulation of upstream snow in absence of snowfall at the PE station only accounts for 35 % of the observed values. It is unclear whether this is due to an underestimation of the mass transport in RACMO2.3, uncertainty in the observations or due to the potential effect of small-scale processes (e.g. sastrugi) in the observational record. A network of
blowing snow sensors would be needed to further address this issue.

Observations for this study were limited to one location over the AIS. As such, results might depend on the local topographical and meteorological conditions. However, as the main conclusions are based on both an analysis of synoptic and local meteorology, deductions of our work are also deemed to be valid at other coastal and escarpment areas of the AIS. Future work should expand the measurements of the individual components of the local SMB to other sites over the AIS, in order to confirm
the role of meteorological conditions at other areas including their effect on the local SMB.

*Data availability.* Data from the instrumentation at the Princess Elisabeth station can be obtained from the database on http://www.aerocloud.be

*Competing interests.* The authors declare that they have no conflict of interest.

*Acknowledgements.* Two anonymous reviewers are acknowledged for their comments significantly improving the manuscript. We thank the logistic teams for executing the yearly maintenance of our instruments at the Princess Elisabeth station and for their help by installing and setting up the new AWS. We thank Wim Boot, Carleen Reijmer, and Michiel van den Broeke (Utrecht University, Institute for Marine and Atmospheric Research Utrecht) for the development of the Automatic Weather Station, technical support and raw data processing. Jan Lenaerts is acknowledged for sharing the data from the RACMO2.3 high-resolution simulation. This work was supported by the Belgian Science Policy Office (BELSPO; grant number BR/143/A2/AEROCLOUD) and the Research Foundation Flanders (FWO; grant number G0C2215N).

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
