# Peer review of "How does the ice sheet surface mass balance relate to snowfall? Insights from a ground-based precipitation radar in East Antarctica"

_The Cryosphere, 2017_

## Referee Comment (RC1) · Anonymous Referee #1 · 31 Jan 2018

The manuscript entitled "How does the ice sheet surface balance relate to snowfall? Insights from a ground-based precipitation radar in East Antarctica" deals with the very important issue of measuring the Surface Mass Balance (SMB) over the Antarctic Ice Sheet. In particular, the goal of this work is to quantify the different terms of the SMB at Princess Elisabeth Station (Antarctica) and investigate the relation between snowfall and accumulation.

The manuscript is for sure within the scope of the Journal and gives a systematic and rigorous analysis of the relation between snowfall and the accumulation at the considered site. Of course this work provides good results but many other sites must

be analyzed to come to a more general conclusion.

Before the manuscript could be published, however, some points should be clarified.

p.3 l.5-9: "Both drifting snow sublimation and surface sublimation have been quantified for the PE station (Thiery et al., 2012). At the local scale, their significance can be fairly large (e.g. King et al., 2001; Bliss et al., 2011; Gorodetskaya et al., 2015; Grazioli et al., 2017b). However, this study mainly focuses on the relation between accumulation / ablation and snowfall. These terms, together with melt which is only relevant at coastal areas and ice shelves (Lenaerts et al., 2017), are therefore only quantified and not investigated in great depth." * "These terms" in this context seem to refer to accumulation/ablation and snowfall. But I guess those are not the terms the authors don't want to investigate in great depth. The authors probably mean "drifting snow sublimation and surface sublimation" terms, but the sentence should be reworded to be more clear.

p.3 l.10: "The total SMB or snow height" * The SMB is usually measured in unity of mass per surface or mm of water equivalent. If the authors want to link the concept of SMB to the snow height, the connection, although comprehensible, needs to be clarified. The same comment is valid for any time the authors talk about SMB height or height changes (p.3 l.19 or l.23 or p.5 l.24 as an example).

p.4 l.11: "This instrument is based on a high speed camera and able to obtain detailed information about snowflake microphysics." * "This instrument is based on a high speed camera and is able to obtain detailed information about snowflake microphysics."

p.4 l.14: "The net local SMB is measured directly by an AWS" * The authors need to explicitly describe or at least mention what kind of instruments are used for a direct measure of SMB. Acoustic height sensors are mentioned but there is no explicit connection with SMB measures.

p.5 l.20: "The local SMB constitutes of the sum of different components (Eq. 1), which can be estimated from measurements using the ground-based instrumentation listed

in Sect. 2.1" * It would be useful here to summarize what ground-based instrument is used to measure each SMB component.

p.6 l.4: "(this corresponds to 1 cm of snow when the density of fresh snow equals 100 kg m−3 )" * The snow height strongly depends on the density of the snowfall particles. In my opinion it is misleading to provide here a general conversion value being the precipitating particles over Antarctica so variable in shape and density.

p.6 l.11-12: "These measurements are processed following Gorodetskaya et al. (2013). Snowfall amounts are obtained from the MRR after applying the Ze-SR relation and methodology determined in Souverijns et al. (2017)." * Please provide here a brief description of Gorodetskaya et al. (2013) processing method and some more information about the Ze-S relationship determined by Souverijns et al. (2017). It is not necessary to provide a full description, but at least give to the reader enough information to be able to go on reading and understanding the methodologies without necessarily reading the reference (in my opinion references should provide a full detailed description of what the authors want to say but the text within the manuscript should be descriptive enough to allow a fluent reading). As an example, the Clear Sky Index methodology is briefly described (l.15-16) even if the proper reference is provided.

p.6 l.30: Again, few words about SANDRA optimization algorithm, why did you choose this one instead of another one? Describe at least the main characteristics that made the authors choose it.

p.7 l.3: Describe the Fast Silhouette Index, mentioning also within the text and not only in the S1 fig. caption that low values are good etc.

p.7 l.21: "that surface and drifting snow sublimation (SUs and SUds respectively) are mainly negative." * Saying that they are mainly negative does it mean that they could be also positive? Fig. 2 caption says that they are plotted as ablation terms. . .

p.7 l.19: "This behaviour is also visible in the ERds" * It should be obvious being the

ERds term just a residual term.

p.9 l.7: Does the "index" have a name? or a reference?

p.11 l.6: "(left side of the graph)" * (left side of the graph - negative pressure difference values)

p.11 l.17-18: . "Accumulation events are characterised by a larger temporal extent of the cloud structure compared to ablation." * It would be interesting here to report the temporal extent of the cloud structure relative to the snowfall event duration because we would expect the persistence of cloud structures at least during the snowfall event. According to AWS, for SMB+ the temporal extent of the cloud structure is 252% of the snowfall event duration, while for SMB- it is 263%, so higher. Opposite trend for ERA-I, with 277% for SMB+ and 243% for SMB-. So, absolutely specking, accumulation events are characterized by a larger temporal extent of the cloud structure, but relatively speaking, some other considerations could be done.

p.12 l.13-14: "From this, the ceilometer was able to detect 486 cm (i.e. the sum of all height changes during events for which blowing snow was detected)" * The ceilometer is not able to detect the snow height, it can detect blowing snow and then the acoustic height sensor can measure the height changes. Please reword the sentence.

p.12 l.16-17: "During all snowfall events, a total amount of 230 mm w.e. (approximately 230 cm in case fresh snow density equals 100 kg m−3) was registered by the MRR, which is lower than the height changes recorded by the AWS (542 cm). This indicates the importance of the continuous movement of snow during snowfall events." * Without any information about snow density, the conversion of 1 mm w.e. to 1 cm cannot be considered realistic. PIP information should be used case by case to convert the w.e. to height and only after comparing the results to AWS measures. On the contrary, if in this work density information are used somehow for the conversion, it should be reported in the manuscript. Without clarifying this point, any conclusion made from this comparison cannot be considered reliable.

p.15 l.14: "The ceilometer was only able to detect 274 cm" * Again, the ceilometer detects the blowing snow, not the snow height.

p.16 l.5: "both accumulation as ablation" * "and" or "as well as"

p.16 l.7: "The distinction between accumulation and ablation events during snowfall was attributed to the duration of the event and the temporal cloud extent" * I would say "was related" or "we found a correlation between..." instead of "was attributed".

Supplement p.3 l.21: "influenced"

Fig.4a: "Wind rose showing the speed and direction of the snowfall events" * "Wind rose showing the speed and direction of the wind during snowfall events"

Fig. 5a: It is difficult to appreciate lines color differences, I would suggest to change the colorscale. Moreover, a legend describing the different lines would be useful.

---

## Referee Comment (RC2) · Anonymous Referee #2 · 1 Feb 2018

Review paper from Souverijns et al. TCD (2018)

This paper presents a compilation of data from a set of instruments (Micro Rain Radar, ceilometer, Automatic Weather Station, among others) over a time period of 37 months, at the Princess Elisabeth (PE) Station, Antarctica. The authors analyse the different situations leading to accumulation and ablation at this site and conclude that "SMB records cannot be considered a good proxy for snowfall at the local scale". The authors compare the typical accumulation and ablation situations to results from a cluster analysis that describes the main weather situations at PE. Results suggest that a large part of the accumulation/ablation at the station takes place after the main precipitation

events because "the fresh snow is easily picked up and transported in shallow drifting snow layers to more inland locations", even when wind speed is relatively low (< 7 ms−1). This latter conclusion is intriguing because it suggests that snow drift may transport snow to more inland locations (here I understand upslope), which is clearly the opposite to present knowledge. Indeed, in Antarctica, it is assumed that snow is mainly drifted downslope due to the occurrence of katabatic winds and that significant amounts of snow are even drifted away from the ice cap to the ocean (Scarchilli et al., 2010; Palm et al., 2017). If the authors' conclusion is true, this process may have large consequences on our vision of snow transport and on accumulation processes in the coast to plateau transition zone. As a consequence, a clear demonstration of this finding is key in the present study, but in the present form, I am not convinced by their demonstration and several verifications are required. In particular, the authors should refer to results from Libois et al. (2014) who performed snow height measurements at Dome C, Antarctica, and demonstrated that the motion of sastrugis during low winds may induce local snow accumulation or erosion at a decameter scale, leading to local snow height variations which seem to be very similar to the accumulation/erosion events presented in the present paper. Libois et al. (2014) suggest that "at each drift event, significant amounts of snow are deposited over approximately 20% of the total area only". This justifies that small amounts of deposited snow may move locally with low winds, and may accumulate in another location nearby, after the end of the precipitation event. If my suggestion is true, then the snow is only moved locally and not transported from (or to) remote areas.

This paper is well written and presents an interesting comparison between different sensors, but key information is lacking and new results (when compared to the previous publications) should be better highlighted: 1. The authors do not sufficiently present the methods, measurements and their uncertainty, and the reader has to refer to previous papers from the same team in order to find the information (van Tricht et al., 2014; Gorodeteskaya et al., 2013, 2015, Souverijns et al., 2017; Gossart et al., 2017). In particular, the sensor uncertainties are not presented, which impedes getting an accurate idea of the quality of the results => Please give more information on sensors and methods in the manuscript.

In particular, estimates are made in mm w.e. but there is no information on snow density at site. What is occurring with density changes caused by snow aging? What is the final uncertainty in ERds estimates? What is the representativeness of one acoustic gauge measurement at decameter or kilometer scale? Is it physically justified to compare local acoustic gauge measurements with radar data that are integrating precipitation amounts at a kilometer scale? Are there any differences between precipitation estimates from Gorodetskaya et al., 2015 (where the ceilometer and the MRR data are combined to analyse precipitation and clouds statistics) and those presented here => please clarify these points

2. Almost all the ideas and conclusions have been presented in Gorodetskaya et al. (2013, 2015). The main interests here are 1) the extent of the dataset which allows to make a statistical analysis on a long time scale, 2) the statistical approach (cluster analysis) made to retrieve the main weather situation at PE. However, this cluster analysis is different from the one presented in previous studies and it leads to a different weather atlas. Please justify the differences. Does the choice of cluster have an impact on present conclusions?

3. The main conclusion (Accumulation and ablation also occur during non-snowfall conditions) is already presented in Gorodetskaya et al. 2015, but the new idea is that snow may be transported from low lying regions => Please refer to Libois et al. (2014) paper, and try to see whether their analysis may help in understanding the snow height changes observed during low winds at the PE station.

4. Several conclusions rely on assumptions made on the blowing snow processes, but these processes are not discussed according to current knowledge based on drifting snow measurements (Li and Pomeroy, 1997; Nishimura and Nemoto 2005; Nishimura and Ishimaru 2012; Scarchilli et al., 2010 ; Libois et al., 2014; Trouviiliez et al., 2014,

Barral et al., 2014, Amory et al., 2015, 2016, 2017; Das et al., 2013). In particular, these publications present important information on the typical threshold wind speeds for snow transport. They also already discuss the impact of snow density, of snow aging, and of the sintering temperature on the threshold wind speeds. Finally, they present key information on the link between the "mean slope in the wind direction" and erosion/deposition processes. This knowledge is important to assess whether large drifting snow fluxes may be observed with low winds for fresh snow. Please refer to these publications and discuss the results accordingly.

5. The situation of the PE station is not discussed in the text, but the station is located at the lee of a rock crest. Are these results site specific? Is there any information on the distribution of accumulation at a kilometer scale around the station (with stakes and acoustic gauges). What is the snow distribution proposed by a regional scale model such as RACMO2 at that site?

To summarize, the dataset is really interesting in particular for model validation and deserves to be published. However, in its present state, this paper is not a sufficiently new contribution compared to (Gorodetskaya et al., 2013, 2015). Before publication in the Cryosphere, I suggest that the authors compare their results to the available out-puts from the RACMO2 regional scale circulation model. This model includes snowdrift processes and outputs may allow "validating" their assumption. If the authors prefer the use of another model, of course, it would make sense. For instance, Stefaan Lhermite is co-author of the present paper and already published a paper using the wind trans-port submodel SnowTran-3D (Groot Zwaaftink et . al. 2011; Gascoin et al., 2013) => Is it possible to use this model here in order to see whether the assumption made on snow transport during low winds is physically supported or not?

As a conclusion, I suggest the authors to make major revisions on their paper. I pro-pose:1) to describe the differences with Gorodetskaya et al. (2013, 2015) and to de-velop the paper around the interest of the new weather atlas presented here, 2) to present more clearly the accuracy of sensors and of their estimates, and (accounting

for the uncertainty of estimations) to critically discuss the variations of the surface level measured with the acoustic gauge, 3) to discuss the potential discrepancies between signals which are describing very different spatial scales, 4) to propose a comparison with a model including drifting snow processes (for instance RACMO2) allowing to give clear evidence on the occurrence of drifting snow events transporting snow from lower altitude areas OR 5) to consider that sastrugis may move locally when the wind is not very strong (Libois et al., 2014). According to their opinion (point 4 or 5), I propose that the authors say whether processes only impact local snow redistribution or impact the regional scale snow redistribution.

Minor comments

Abstract ilne 8: please remove "an unprecedented"

Lines 9-11:The authors write: "However, snowfall events are much more common than accumulation events. During 38% of the snowfall cases observed, the freshly-fallen snow is ablated by the wind during the course of the event. Generally, snow storms of longer duration have a higher chance to attain for accumulation at the local scale, while shorter events usually attain for ablation" This conclusion is very similar to the following one given in Gorodetskaya et al., 2015: "Large accumulation events (> 10mmw.e. day-1) during the radar-measurement period of 26 months were always associated with snowfall, but at the same time other snowfall events did not always lead to accumulation.". Please clarify which are the new conclusions in the present paper.

Line 12: "As such, SMB records cannot be considered a good proxy for snowfall at the local scale." => Considering the decameter to kilometer scale variability of the SMB, or alternating of megadunes/wind glaze' areas for instance, this conclusion is trivial. Please reformulate.

Lines 13-15: "when preceding snowfall events were occurring in upstream coastal areas. This fresh snow is easily picked up and transported in shallow drifting snow layers to more inland locations, even when wind speed is relatively low (< 7 ms−1)." =>

please refer to (Libois et al., 2014) where a potential explanation of such variations without precipitation is proposed. If the authors do not concur with these explanations, please demonstrate clearly that the snow can be transported upslope.

Introduction: Page 1 - Line 20 "an important regulator of the present and future global climate and water cycle" => I don't understand this sentence. The oceans are currently a regulator since they absorb 93% of the global warming, but what do the authors mean with the word "regulator" in the case of Antarctica?

Page 1 – Line 22: what do you mean with coupled climate/surface models? Do you mean GCM? Coupled AOGCM? AOGCM coupled with dynamic changes in ice surfaces. The cited publications are referring to very different models and do not clearly indicate what kind of model the authors are referring to.

Page 2-Line 5: please refer to papers from Grazioli et al. (2017a&b) because these paper also offer new information on precipitation measurements in Antarctica.

Page 2, Line 8: the authors cite: (Vaughan et al., 1999 and Magand et al., 2007). However, the database from Vaughan et al. (1999) has been updated by Favier et al. (2013) and Wang et al. (2016).

Page 2 Line 9: "Gorodetskaya et al. (2015) were the first to quantify the different terms" => "Gorodetskaya et al. (2015) quantified the different terms"

Equation 1: I do not understand how "Sublimation of the drifting snow" and ERds may be separated? if a snow flake is sublimated while it is drifted, this means that it has been eroded first. Is the mass loss accounted twice in the calculation? i.e., is it accounted for (1) in sublimation and (2) in ERds? please explain this formulation

Page 2, line 18: stake measurements and ice cores => please refer to (Thomas et al., 2017)

Page 2, line 19: "the separate measurement of any of the components of the local SMB is considered a difficult task" => please also refer to (Eisen et al., 2008 ; Gorodetskaya

et al., 2015; Amory et al., 2017 ; Grazioli et al., 2017)

Page 2, line 31: "Wind-induced accumulation / ablation by drifting / blowing snow over the AIS has an important impact on the local SMB" => please refer to (Palm et al., 2017)

Page 2, line 34: "using a network of snowdrift instrumentation (Leonard et al., 2011)" => please also refer to (Nishimura and Nemoto 2005; Nishimura and Ishimaru 2012; Trouvilliez et al., 2014 ; Amory et al., 2017)

Page 2, line 35: "Neglecting this term might however lead to significant errors" => How much?

Page 3, line 10: "The total SMB or snow height can be measured by an Automatic Weather Station (AWS)," => this sentence is not correct because AWS do not offer any information on snow density. More generally there is no information on snow density in the present paper. Do the authors consider the same value of density all the time? How do the authors consider variations in density related to snow aging? Please reformulate and clearly describe how the snow density is considered in calculations.

Page 3, line 19: "noted that snowfall events at the PE station do not necessarily contribute to accumulation or an increase in the height of the local SMB." => The authors write that their main conclusions have already been given in previous publications. Please describe what the new insights are? For instance, the present study may offer a more robust estimation of the frequency of different events.

Section 2.1: page 3, line 28: "In order to gain insight in the relation between snowfall and the SMB, reliable, high-frequency and long-term in situ observations are indispensable."=> repetition with the introduction. this sentence may be removed

Figure 1: the P-E station appears to be close to a blue ice area (located southward) and in the lee of a mountain ridge. Is there any impact of the location on local accumulation? For instance, when the wind is coming from the south, it is coming from the

blue ice area and the ridge, which are characterized by erosion: snow is not available for transport, but these areas are in the vicinity of the PE station. Conversely, the areas located close to the PE station but in the ENE direction are covered by snow. Erosion and transport from these areas thus occur more easily, attaining for deposition at the PE station. This may suggest that snow is transported only over small spatial scales when the wind speed is low and not necessarily from further low-lying areas, in agreement with the processes described by Libois et al. (2014). As a consequence, the impact on accumulation distribution may be only local. What is the authors' opinion?

Page 4, line 13: "snowfall rates was obtained including a constrain on its uncertainty (Souverijns et al., 2017)." => please give uncertainty values for all the sensors.

Page 4, line 16: "Furthermore, it is equipped by an acoustic height sensor, which is able to measure snow height changes with an accuracy of 1 cm on an hourly time resolution." and later, "resolution. Snowfall events are defined as exceeding the threshold of 1 mm w.e. during the continuous duration of snowfall measured by the MRR (this corresponds to 1 cm of snow when the density of fresh snow equals 100 kg m$-3$)." =>(1) This threshold may be too high to depict feeble precipitation: do results depend on this threshold? =>(2) In (Libois et al., 2014), they write:"we chose to set the density of fresh snow to 170 kg m$-3$, which corresponds to the fifth lowest percentile of the measured surface densities at Dome C during the 2012–2013 campaign.". In this study at Dome C, where diamond dust and surface hoar may occur, the fresh snow density is already 70% higher than at PE. Please justify the choice of this very low snow density value (100kg m-3). Is there any impact of this choice on the comparison between MRR precipitation and surface level variations? If higher density values are considered, then the 1cm threshold would lead to neglect events with higher accumulation?

page 5, line 4: "Recently, an algorithm for the detection of blowing snow by the use of a ceilometer was developed"=> in (Gorodetskaya et al., 2015), a combined analysis of the ceilometer and of the MRR is done to get information on clouds and precipitation, here the ceilometer is used to assess blowing snow events? Here, do the authors use

the ceilometer to also analyse the clouds characteristics?

page 5, line 8: "The minimal height of the blowing snow layer to be detected by the ceilometer equals 30 m at the PE station."=> please give sensor uncertainty and the impact on the blowing snow flux estimate.

page 5, line 9: "Surface and snowdrift sublimation are quantified using the approach of Thiery et al. (2012)" => please give uncertainty values

page 5, line 18: "which is unprecedented for the Antarctic region."=> There are many other similar datasets in Antarctica available for many years but with different sensors and focus (e.g. the AWS network). Please remove or reformulate.

Page 5, line 22: "Nevertheless, since the AWS measures the total SMB directly, ERds can be calculated as the residual term after inverting Eq. 1" => this calculation indicates that ERds values include the sum of all other uncertainties. What is the accuracy of this term? Moreover, the different sensors used here refer to different spatial scales (km^2 scale for the MRR and ceilometer, but a few m^2 for the ultrasonic gauges). Is it physically justified to directly compare these very different scales? In order to accurately compare the MRR/ceilometer data with surface height data, it may be necessary to consider snow height variations given by multiple sensors distributed over the area. In particular, referring to Libois et al. (2014) publication, is it possible that snow accumulates in depressions or along small snow barchans, which may move due to snow reptation? For instance, please observe the 3 large increases and decreases of the surface level in January, February and March 2012 (Figure 2).

Section 2.3 Page 6, Line 12: give more details on the method, and particularly on measurement uncertainty.

Page 6, line 30: "The SANDRA optimisation algorithm was selected to perform this cluster analysis," => Please explain the interest of this new cluster analysis. Why is it more interesting and robust than the one used in Gorodetskaya et al. 2013? The

weather atlas presented in the two different papers looks different, please justify these differences.

Page 7, lines 3-6: "a total of nine circulation patterns was selected. [. . .] during the four types of events attaining for a change in the local SMB" => in figures, 6 patterns are presented. Please clarify. Which are the four main patterns within the 6 patterns?

Page 7, line 13: "are converted to water equivalent values"=> please indicate how density changes are considered in the first snow layers?

Page 8 Figure 2: This figure is similar to Figure 9 from (Gorodetskaya et al., 2015). I suggest the authors to present their results over the whole 37 months

Page 8 Line 5: did the authors compare Precipitation estimates from ERA-Interim with precipitation from the MRR? is it possible to consider that interpretation of the cluster and backtrajectory analyses are robust If ERA precipitations do not fit with field measurements?

Section 3.3 Page 8, line 10: "in total 50 independent snowfall episodes were detected attaining for at least 1 mm w.e. at the PE station" => Libois et al. (2014) present a statistical analysis and write that: "Under the ergodic hypothesis, we conclude that at each drift event, significant amounts of snow are deposited over approximately 20% of the total area only". This suggests that there is only a low probability to capture snow accumulation during drifting snow events with only one acoustic gauge. Do the authors believe that their results may change if they had access to the mean variations given by 3 or 5 sonic gauges separated by 10m from each other?

Page 9: Figure 3. Cases 3 and 4 look very similar. Please indicate more clearly in the text and in captions which are the Cases C1, C2, etc. . . This interpretation of the cluster should be strengthened in the text and compared with previous papers. Do the conclusions depend on the clusters analysis?

Page 11, line 12: "From the total number of snowfall events, 31 resulted in accumulation

(62 %), while 19 led to ablation (38 %)." => is there any relationship with precipitation intensities or amounts given by the MRR data and accumulation/ablation occurrences?

Page 12, line 5: please discuss this paragraph considering conclusions from Libois et al. (2014)

Page 12, line 16:"During all snowfall events, a total amount of 230 mm w.e. (approximately 230 cm in case fresh snow density equals 100 kg m$-3$) was registered by the MRR, which is lower than the height changes recorded by the AWS (542 cm)." => what is the accuracy of both sensors? please also discuss the impact of snow density value on results. Is it possible to consider that the AWS is located in a small overaccumulation zone (due to gravity waves at the lee of the mountain crest for instance)? I propose the authors to discuss these differences.

Page 13, line 1: "Accumulation records are therefore not advised to be used as a proxy for snowfall over East Antarctica." => please, replace "accumulation" by "punctual accumulation". Indeed, I suppose that the conclusion would be different if the authors had access to many stake/ice cores/sonic gauge data over a large area.

Section 3.4 Page 13, line 6:"snow displacement events also occur without the presence of snowfall." => what is horizontal scale here? Deca-centimeter? Kilometer? Deca-kilometer?

Page 14, Line 2:"limiting the regions that receive snowfall to coastal areas" => how do the authors observe that precipitation occur in coastal areas? Do they use ERA-interim outputs?

Page 14, line3: "to be limited to the coastal areas, not reaching the PE station" => I suppose that the MRR data are used in order to reach this conclusion. However, Gorodetskaya et al., (2015) write:"While MRR misses the most feeble precipitation (virga or snowfall) with Ze < -8 dBz, its sensitivity is sufficient to detect typical precipitation at the site. [. . .] and ice clouds or weak precipitation not detected by MRR (22%

of the total period or 63% of the overcast)." => I suppose that feeble precipitations are not always detected by the MRR. If it were the case, the MRR would not be sufficient to demonstrate that no precipitation occurred. Please clarify.

Page 14, line 16: "A fraction of 46 % of the ablation events take place within 24 hours after a snowfall event. As stated in Sect. 3.3, snowfall events are mainly characterised by a ENE flow (Fig. 4a)."=> do the authors estimate the "reptation" velocity (e. g., Nishimura et al., 2014) in order to estimate the distance over which the snow particles were transported until reaching PE station?

Page 15, line 8: "the occurrence of accumulation / ablation at very low wind speeds (Fig. 7). This points out that the time since the last snowfall event and the amount of low-density fresh snow that is available is of much higher importance than the wind speed in order to attain for blowing snow" => this conclusion is intriguing if we consider that friction velocity has to exceed a threshold friction velocity in order to allow wind drift occurrence. If the wind speed is too low, snow saltation is not possible (e. g., Nishimura and Nemoto, 2005). Please discuss this point considering current knowledge on blowing snow processes (e. g., Nishimura and Nemoto, 2005).

Page 15, lines 15-20: "A total of 1125 cm in SMB changes was measured by the AWS during the events. The ceilometer was only able to detect 274 cm, indicating that the blowing snow layer only has a limited vertical extent and that the transport of snow is restricted to shallow layers close to the ground during these type of events." => I suppose that blowing snow events with a limited vertical extent are transporting weak amounts of snow. How can this type of events explain the difference between 1125 cm and 274 cm?

Furthermore, it is noted that "ablation and accumulation can significantly compensate each other" => please discuss this paragraph considering Libois et al. (2014) results.

Page 15, line 10: please cite (Amory et al., 2017).
Page 15, line 25: please cite (Nishimura and Ishimaru 2012 ; Trouvilliez et al. 2014)

Page 15: the first paragraph of the conclusion is not really necessary (repetition of the introduction)

Page 16: "Meteorological conditions during snowfall, accumulation and ablation, were indicated, including their impact on the local SMB, which was largely unknown up to now." => Gorodetskaya et al. (2015) already proposed estimates of each SMB component. Please clarify the differences between both estimates, and justify why the present estimate is more accurate than the one proposed by Gorodetskaya et al. (2015).

In the references: (Leonard et al., 2011) => (Leonard et al., 2012) Please include missing information in (Stohl et al., 1995)

References

Amory, C., Gallée, H., Naaim-Bouvet, F., Favier, V., Vignon, E., Picard, G., Trouvilliez, A., Piard, L., Genthon, C., Bellot, H., 2016. Seasonal Variations in Drag Coefficient over a Sastrugi-Covered Snowfield in Coastal East Antarctica. Bound.-Layer Meteorol., 164(1), 107–133, https://doi.org/10.1007/s10546-017-0242-5

Amory, C., Trouvilliez, A., Gallée, H., Favier, V., Naaim-Bouvet, F., Genthon, C., Agosta, C., Piard, L., Bellot, H., 2015. Comparison between observed and simulated aeolian snow mass fluxes in Adélie Land, East Antarctica. The Cryosphere 9, 1373–1383. https://doi.org/10.5194/tc-9-1373-2015

Barral, H., Genthon, C., Trouvilliez, A., Brun, C., Amory, C., 2014. Blowing snow in coastal Adélie Land, Antarctica: three atmospheric-moisture issues. The Cryosphere 8, 1905–1919. https://doi.org/10.5194/tc-8-1905-2014

Das, I., Bell, R.E., Scambos, T.A., Wolovick, M., Creyts, T.T., Studinger, M., Frearson, N., Nicolas, J.P., Lenaerts, J.T.M., van den Broeke, M.R., 2013. Influence of persistent wind scour on the surface mass balance of Antarctica. Nat. Geosci. 6, 367–371. https://doi.org/10.1038/ngeo1766

Eisen, O., Frezzotti, M., Genthon, C., Isaksson, E., Magand, O., van den Broeke, M.R., Dixon, D.A., Ekaykin, A., Holmlund, P., Kameda, T., Karlöf, L., Kaspari, S., Lipenkov, V.Y., Oerter, H., Takahashi, S., Vaughan, D.G., 2008. Ground-based measurements of spatial and temporal variability of snow accumulation in East Antarctica. Rev. Geophys. 46, RG2001. https://doi.org/10.1029/2006RG000218

Favier, V., Agosta, C., Parouty, S., Durand, G., Delaygue, G., Gallée, H., Drouet, A.-S., Trouvilliez, A., Krinner, G., 2013. An updated and quality controlled surface mass balance dataset for Antarctica. The Cryosphere 7, 583–597. https://doi.org/10.5194/tc-7-583-2013

Gascoin, S., Lhermitte, S., Kinnard, C., Bortels, K., Liston, G.E., 2013. Wind effects on snow cover in Pascua-Lama, Dry Andes of Chile. Adv. Water Resour., Snow–Atmosphere Interactions and Hydrological Consequences 55, 25–39. https://doi.org/10.1016/j.advwatres.2012.11.013

Gorodetskaya, I.V., Kneifel, S., Maahn, M., Thiery, W., Schween, J.H., Mangold, A., Crewell, S., Van Lipzig, N.P.M., 2015. Cloud and precipitation properties from ground-based remote-sensing instruments in East Antarctica. The Cryosphere 9, 285–304. https://doi.org/10.5194/tc-9-285-2015

Gorodetskaya, I.V., Van Lipzig, N.P.M., Van den Broeke, M.R., Mangold, A., Boot, W., Reijmer, C.H., 2013. Meteorological regimes and accumulation patterns at Utsteinen, Dronning Maud Land, East Antarctica: Analysis of two contrasting years. J. Geophys. Res. Atmospheres 118, 1700–1715. https://doi.org/10.1002/jgrd.50177

Gossart, A., Souverijns, N., Gorodetskaya, I.V., Lhermitte, S., Lenaerts, J.T.M., Schween, J.H., Mangold, A., Laffineur, Q., van Lipzig, N.P.M., 2017. Blowing snow detection from ground-based ceilometers: application to East Antarctica. The Cryosphere 11, 2755–2772. https://doi.org/10.5194/tc-11-2755-2017

Grazioli, J., Genthon, C., Boudevillain, B., Duran-Alarcon, C., Del Guasta,

M., Madeleine, J.-B., Berne, A., 2017. Measurements of precipitation in Dumont d'Urville, Adélie Land, East Antarctica. The Cryosphere 11, 1797–1811. https://doi.org/10.5194/tc-11-1797-2017

Grazioli, J., Madeleine, J.-B., Gallée, H., Forbes, R.M., Genthon, C., Krinner, G., Berne, A., 2017. Katabatic winds diminish precipitation contribution to the Antarctic ice mass balance. Proc. Natl. Acad. Sci. 114, 10858–10863. https://doi.org/10.1073/pnas.1707633114

Groot Zwaaftink, C.D., Cagnati, A., Crepaz, A., Fierz, C., Macelloni, G., Valt, M., Lehning, M., 2013. Event-driven deposition of snow on the Antarctic Plateau: analyzing field measurements with SNOWPACK. The Cryosphere 7, 333–347. https://doi.org/10.5194/tc-7-333-2013

Leonard, K.C., Tremblay, L.-B., Thom, J.E., MacAyeal, D.R., 2012. Drifting snow threshold measurements near McMurdo station, Antarctica: A sensor comparison study. Cold Reg. Sci. Technol. 70, 71–80. https://doi.org/10.1016/j.coldregions.2011.08.001

Li, L., Pomeroy, J.W., 1997. Estimates of threshold wind speeds for snow transport using meteorological data. J. Appl. Meteorol. 36, 205–213.

Libois, Q., Picard, G., Arnaud, L., Morin, S., Brun, E., 2014. Modeling the impact of snow drift on the decameter-scale variability of snow properties on the Antarctic Plateau. J. Geophys. Res. Atmospheres 119, 11,662–11,681. https://doi.org/10.1002/2014JD022361

Magand, O., Genthon, C., Fily, M., Krinner, G., Picard, G., Frezzotti, M., Ekaykin, A.A., 2007. An up-to-date quality-controlled surface mass balance data set for the 90°–180°E Antarctica sector and 1950–2005 period. J. Geophys. Res. Atmospheres 112, D12106. https://doi.org/10.1029/2006JD007691

Nishimura, K., Ishimaru, T., 2012. Development of an automatic blowing-snow station.

Cold Reg. Sci. Technol. 82, 30–35. https://doi.org/10.1016/j.coldregions.2012.05.005

Nishimura, K., Nemoto, M., 2005. Blowing snow at Mizuho station, Antarctica. Philos. Trans. R. Soc. Math. Phys. Eng. Sci. 363, 1647–1662. https://doi.org/10.1098/rsta.2005.1599

Palm, S.P., Kayetha, V., Yang, Y., Pauly, R., 2017. Blowing snow sublimation and transport over Antarctica from 11 years of CALIPSO observations. The Cryosphere 11, 2555–2569. https://doi.org/10.5194/tc-11-2555-2017

Scarchilli, C., Frezzotti, M., Grigioni, P., De Silvestri, L., Agnoletto, L., Dolci, S., 2010. Extraordinary blowing snow transport events in East Antarctica. Clim. Dyn. 34, 1195–1206. https://doi.org/10.1007/s00382-009-0601-0

Souverijns, N., Gossart, A., Lhermitte, S., Gorodetskaya, I.V., Kneifel, S., Maahn, M., Bliven, F.L., van Lipzig, N.P.M., 2017. Estimating radar reflectivity - Snowfall rate relationships and their uncertainties over Antarctica by combining disdrometer and radar observations. Atmospheric Res. 196, 211–223. https://doi.org/10.1016/j.atmosres.2017.06.001

Stohl, A., Wotawa, G., Seibert, P., Kromp-Kolb, H., 1995. Interpolation Errors in Wind Fields as a Function of Spatial and Temporal Resolution and Their Impact on Different Types of Kinematic Trajectories. J. Appl. Meteorol. 34, 2149–2165. https://doi.org/10.1175/1520-0450(1995)034<2149:IEIWFA>2.0.CO;2

Thiery, W., Gorodetskaya, I.V., Bintanja, R., Van Lipzig, N.P.M., Van den Broeke, M.R., Reijmer, C.H., Kuipers Munneke, P., 2012. Surface and snowdrift sublimation at Princess Elisabeth station, East Antarctica. The Cryosphere 6, 841–857. https://doi.org/10.5194/tc-6-841-2012

Thomas, E.R., van Wessem, J.M., Roberts, J., Isaksson, E., Schlosser, E., Fudge, T.J., Vallelonga, P., Medley, B., Lenaerts, J., Bertler, N., van den Broeke, M.R., Dixon, D.A., Frezzotti, M., Stenni, B., Curran, M., Ekaykin, A.A., 2017. Regional

Antarctic snow accumulation over the past 1000 years. Clim. Past 13, 1491–1513. https://doi.org/10.5194/cp-13-1491-2017

Trouvilliez, A., Naaim-Bouvet, F., Genthon, C., Piard, L., Favier, V., Bellot, H., Agosta, C., Palerme, C., Amory, C., Gallée, H., 2014. A novel experimental study of aeolian snow transport in Adelie Land (Antarctica). Cold Reg. Sci. Technol. 108, 125–138. https://doi.org/10.1016/j.coldregions.2014.09.005

Van Tricht, K., Gorodetskaya, I.V., Lhermitte, S., Turner, D.D., Schween, J.H., Van Lipzig, N.P.M., 2014. An improved algorithm for polar cloud-base detection by ceilometer over the ice sheets. Atmospheric Meas. Tech. 7, 1153–1167. https://doi.org/10.5194/amt-7-1153-2014

Vaughan, D.G., Bamber, J.L., Giovinetto, M.B., Russell, J., Cooper, A.P.R., 1999. Reassessment of Net Surface Mass Balance in Antarctica. J. Clim. 12, 933–946. https://doi.org/10.1175/1520-0442(1999)012<0933:RONSMB>2.0.CO;2

Wang, Y., Ding, M., van Wessem, J.M., Schlosser, E., Altnau, S., van den Broeke, M.R., Lenaerts, J.T.M., Thomas, E.R., Isaksson, E., Wang, J., Sun, W., 2016. A Comparison of Antarctic Ice Sheet Surface Mass Balance from Atmospheric Climate Models and In Situ Observations. J. Clim. 29, 5317–5337. https://doi.org/10.1175/JCLI-D-15-0642.1

---

## Author Comment (AC1) · 22 Mar 2018

**Response to Reviewer 1 Comments:**
**How does the ice sheet surface mass balance relate to snowfall? Insights from a ground-based precipitation radar in East Antarctica**

Niels Souverijns, Alexandra Gossart, Irina V. Gorodetskaya, Stef Lhermitte, Alexander Mangold, Quentin Laffineur, Andy Delcloo, Nicole P.M. van Lipzig

March 22, 2018

For clarifying our answers to the reviewers' comments, the following color scheme is used: comments of the reviewer are denoted in blue, our answers are denoted in black and quotes from the revised text are in green.

*1. The manuscript entitled "How does the ice sheet surface balance relate to snowfall? Insights from a ground-based precipitation radar in East Antarctica" deals with the very important issue of measuring the Surface Mass Balance (SMB) over the Antarctic Ice Sheet. In particular, the goal of this work is to quantify the different terms of the SMB at Princess Elisabeth Station (Antarctica) and investigate the relation between snowfall and accumulation. The manuscript is for sure within the scope of the Journal and gives a systematic and rigorous analysis of the relation between snowfall and the accumulation at the considered site. Of course this work provides good results but many other sites must be analyzed to come to a more general conclusion.*

The reviewer is thanked for his comments regarding the manuscript. As noted, the work is only based on results from one site over the Antarctic Ice Sheet and takes advantage of a unique set of instruments, including a precipitation radar and an optical disdrometer, unavailable at other Antarctic stations. We stressed this a bit more clearly in the last section of the conclusion.

*Observations for this study were limited to one location over the AIS. As such, results might depend on the local topographical and meteorological conditions. However, as the main conclusions are based on both an analysis of synoptic and local meteorology, deductions of our work are also deemed to be valid at other coastal and escarpment areas of the AIS. Future work should expand the measurements of the individual components of the local SMB to other sites over the AIS, in order to confirm the role of meteorological conditions at other areas including their effect on the local SMB.*

*2. p.3 l.5-9: "Both drifting snow sublimation and surface sublimation have been quantified for the PE station (Thiery et al., 2012). At the local scale, their significance can be fairly large (e.g. King et al., 2001; Bliss et al., 2011; Gorodetskaya et al., 2015; Grazioli et al., 2017). However, this study mainly focuses on the relation between accumulation / ablation and snowfall. These terms, to-*

*gether with melt which is only relevant at coastal areas and ice shelves (Lenaerts et al., 2017), are therefore only quantified and not investigated in great depth." * "These terms" in this context seem to refer to accumulation/ablation and snowfall. But I guess those are not the terms the authors don't want to investigate in great depth. The authors probably mean "drifting snow sublimation and surface sublimation" terms, but the sentence should be reworded to be more clear.*

The paragraph has been rewritten accordingly. A clear reference to the sublimation terms is added and a sentence was added quantifying the mean annual effect of the sublimation terms over the PE station.

*Both surface and drifting snow sublimation have been quantified for the PE station (Thiery et al., 2012). At the local scale, the significance of the processes can be fairly large (e.g. King et al., 2001; Bliss et al., 2011; Gorodetskaya et al., 2015; Grazioli et al., 2017). For the PE station, sublimation was found to remove 10 % of the total precipitation (Thiery et al., 2012). In this study, the focus is mainly on the relation between accumulation / ablation and snowfall. The sublimation terms, together with melt which is only relevant at coastal areas and ice shelves (Lenaerts et al., 2017), are therefore only quantified and not investigated in great depth.*

*3. p.3 l.10: "The total SMB or snow height" * The SMB is usually measured in unity of mass per surface or mm of water equivalent. If the authors want to link the concept of SMB to the snow height, the connection, although comprehensible, needs to be clarified. The same comment is valid for any time the authors talk about SMB height or height changes (p.3 l.19 or l.23 or p.5 l.24 as an example).*

We agree this concept has not been explained well. The Automatic Weather Station (AWS) at the PE station is equipped with an acoustic height sensor, which is capable of measuring snow height changes. In our study, we consider the SMB in units of millimeter water equivalent. In order to convert the height changes measured by the AWS to millimeters of water equivalent, density measurements of snow need to be available. Every summer season, snow density profile measurements are performed in a 1 m snow pit in the close vicinity of the AWS. Each year, the yearly average density of the snow that accumulated since the past year is calculated. Using this density, the height changes measured by the AWS can be converted to water equivalent. This concept is now explained in the introduction and is reformulated at several locations throughout the manuscript, including the caption of Figure 2.

*Each year, the average density of the snow that has accumulated in the past year is calculated from snow pit measurements at the PE station. Using these average yearly densities (varying between 309-375 kg m$^{-3}$), snow height changes are converted to changes in the local SMB (water equivalent) (Gorodetskaya et al., 2013, 2015).*

*Gorodetskaya et al. (2015) showed that snowfall events at the PE station do not necessarily contribute to accumulation or an increase in snow height.*

*In addition, changes in the local SMB take place without snowfall.*

*The net local SMB is calculated based on snow height measurements of an AWS and yearly snow density records (see Section 1).*

*Nevertheless, since the total SMB can be deduced from the AWS (see Section 1), $ER_{ds}$ can be calculated as the residual term after inverting Eq. 1 (Gorodetskaya et al., 2015).*

*One would therefore expect that snowfall results in accumulation or an increase in height.*

**4. p.4 l.11: "This instrument is based on a high speed camera and able to obtain detailed information about snowflake microphysics." * "This instrument is based on a high speed camera and is able to obtain detailed information about snowflake microphysics."**

The sentence has been adapted accordingly.

*This instrument is based on a high speed camera and is able to obtain detailed information about snowflake microphysics (i.e. the particle size distribution).*

**5. p.4 l.14: "The net local SMB is measured directly by an AWS" * The authors need to explicitly describe or at least mention what kind of instruments are used for a direct measure of SMB. Acoustic height sensors are mentioned but there is no explicit connection with SMB measures.**

This comment has been addressed in comment 3. There, we also expanded the description of the instrumentation and methodology to obtain SMB records from AWS snow height measurements. The sentence is repeated here for clarity. We refer also to the material and methods section, where the full approach is discussed.

*The net local SMB is calculated based on snow height measurements of an AWS and yearly snow density records (see Section 1).*

**6. p.5 l.20: "The local SMB constitutes of the sum of different components (Eq. 1), which can be estimated from measurements using the ground-based instrumentation listed in Sect. 2.1" * It would be useful here to summarize what ground-based instrument is used to measure each SMB component.**

An overview of the different components is added including the instrumentation that is used to measure each of the components. For a detailed description, we referred to the above sections.

*The local SMB is composed of the sum of different components (Eq. 1). Snowfall amounts are measured by a Micro Rain Radar, while the two sublimation terms are calculated based on meteorological data from the AWS (see Section 2.1). The $ER_{ds}$ term is not measured directly at the PE station. Nevertheless, since the total SMB can be deduced from the AWS (see Section 1), $ER_{ds}$ can be calculated as the residual term after inverting Eq. 1 (Gorodetskaya et al., 2015). Its uncertainty is therefore based on the uncertainty of all other components of the local SMB (Eq. 1; Gorodetskaya et al., 2015) and is mainly determined by the uncertainty of the snowfall component at the PE station.*

**7. p.6 l.4: "(this corresponds to 1 cm of snow when the density of fresh snow equals 100 kg m$^{-3}$)" * The snow height strongly depends on the density of the snowfall particles. In my opinion it is misleading to provide here a general conversion value being the precipitating particles over Antarctica so variable in shape and density.**

The conversion of snowfall amounts to height changes is indeed inappropriate and has no added value in this part of the text. It has been removed from this section. See also comment 17 for a clarification of this conversion further in the manuscript.

*8. p.6 l.11-12: "These measurements are processed following Gorodetskaya et al. (2013). Snowfall amounts are obtained from the MRR after applying the Ze-SR relation and methodology determined in Souverijns et al. (2017)." * Please provide here a brief description of Gorodetskaya et al. (2013) processing method and some more information about the Ze-S relationship determined by Souverijns et al. (2017). It is not necessary to provide a full description, but at least give to the reader enough information to be able to go on reading and understanding the methodologies without necessarily reading the reference (in my opinion references should provide a full detailed description of what the authors want to say but the text within the manuscript should be descriptive enough to allow a fluent reading). As an example, the Clear Sky Index methodology is briefly described (l.15-16) even if the proper reference is provided.*

A description of the processing methods of the measurements is added in the section describing the AWS and the MRR / disdrometer.

*The AWS records meteorological variables, such as air temperature, pressure, wind speed and direction, relative humidity and radiative fluxes at 2 meters above the surface. These measurements are processed following Gorodetskaya et al. (2013). Wind speed and directions are recorded by an anemometer at the top of the AWS. Humidity is recorded with respect to water at the top of the AWS and is converted to humidity with respect to ice using the conversion of Anderson (1994). Broadband radiative fluxes are measured using pyranometers and pyrgeometers. All the above parameters are measured with 6-minute resolution and averaged to hourly means. Furthermore, the AWS is equipped with an acoustic height sensor, which measures snow height changes once an hour with an accuracy of 1 cm on an hourly time resolution. A running mean of 24 hours is applied to erase short-term decameter scale variability due to sastrugi movement (Libois et al., 2014). Furthermore, several corrections to remove erroneous data are executed (Gorodetskaya et al., 2011). Apart from this, temporary peaks in the snow height records are excluded from the analysis. In January 2016, a new AWS was set up to replace the one installed in 2009, able to measure snow height changes more accurately. A detailed overview of the specifications of the old AWS including its uncertainty can be found in Gorodetskaya et al. (2013) and of the new AWS in the Supplement (Table S1).*

*In order to obtain reliable estimates of snowfall rates and their uncertainty, an optical disdrometer (Precipitation Imaging Package Newman et al., 2009) was installed at the PE station. This instrument is based on a high speed camera and is able to obtain detailed information about snowflake microphysics (i.e. the particle size distribution). A correction for the horizontal and vertical displacement of snowfall between the MRR data acquisition level and the surface has been applied (Wood, 2011; Souverijns et al., 2017). Using this information, a relation between radar reflectivity measured by the MRR and snowfall rates was obtained: $Ze=18SR^{1.1}$. Furthermore, a constrain on the uncertainty of the resulting snowfall rates was obtained [-59 % +60 %] (10th-90th percentile) (Souverijns et al., 2017), which is a considerable reduction compared to earlier snowfall rate estimates at the PE station that were retrieved without any information on the snow particle microphysical characteristics (Gorodetskaya et al., 2015).*

**9. p.6 l.30: Again, few words about SANDRA optimization algorithm, why did you choose this one instead of another one? Describe at least the main characteristics that made the authors choose it.**

A description was added to the SANDRA algorithm including a justification of the choice for this algorithm.

*Several algorithms are available to perform a cluster analysis (Philipp et al., 2010). In recent studies, thorough evaluations were performed for each of these algorithms, indicating the best performance regarding circulation clustering for optimisation algorithms over different parts of the world (Huth et al., 2008; Beck and Philipp, 2010; Casado et al., 2010; Souverijns et al., 2016). From these optimisation algorithms, the simulated annealing and diversified randomisation (SANDRA) algorithm was chosen, which is based on k-means clustering (Philipp et al., 2007) as it performed adequately for different applications over the world regarding circulation clustering (Huth et al., 2008; Beck and Philipp, 2010; Casado et al., 2010; Souverijns et al., 2016).*

**10. p.7 l.3: Describe the Fast Silhouette Index, mentioning also within the text and not only in the S1 fig. caption that low values are good etc.**

The sentence has been adapted accordingly.

*Using the Fast Silhouette Index, the ability of the SANDRA algorithm to maximise the separability between the members of different circulation patterns, while minimising the variances within each circulation pattern, was investigated (Rousseeuw, 1987). In this study, a total of six circulation patterns was selected. The Fast Silhouette index indicates a local minimum value as a further increase in the total number of circulation patterns shows no significant improvement (i.e. decrease) regarding the classification skill (Fig. S2).*

**11. p.7 l.21: "that surface and drifting snow sublimation (SUs and SUds respectively) are mainly negative." * Saying that they are mainly negative does it mean that they could be also positive? Fig. 2 caption says that they are plotted as ablation terms...**

This sentence have been adapted in order to not result in confusion with the caption in Fig. 2.

*Furthermore, it can also be noted that surface and drifting snow sublimation ($SU_s$ and $SU_{ds}$ respectively) are both ablation terms.*

**12. p.7 l.19: "This behaviour is also visible in the ERds" * It should be obvious being the ERds term just a residual term.**

This part of the sentence has been removed.

*These events occur both with or without snowfall and allows for easy identification of the individual accumulation and ablation events.*

**13. p.9 l.7: Does the "index" have a name? or a reference?**

The index is based on the pressure difference between the PE station and a location northwest of the station where low pressure systems are present attributing for precipitation at the PE station. The location northwest of the station is determined from the results of the cluster analysis (Fig. 3). From the cluster analysis, it was deduced that the higher the pressure difference between both locations, the higher the transport capacity of the station (this can also been seen in Fig. 5b). As this is a site-specific index, it does not have a name or reference. We modified the text so it is emphasised that this index was created by the authors and added a green dot on Fig. 3 in order to denote the locations of the cyclone.

*The transport capacity of the cyclone is parametrised by a self-constructed index based on the pressure difference between the PE station and the typical location of the trough northwest of the station (0° E, 62° S, Fig. 3) as these cyclones attribute for the highest snowfall amounts at the PE station.*

**14. p.11 l.6: "(left side of the graph)" * (left side of the graph - negative pressure difference values)**

The text has been adapted accordingly.

*In case no cyclone is present NW of the PE station (i.e. negative pressure difference values; left side of the graph), snowfall amounts are generally low.*

**15. p.11 l.17-18: "Accumulation events are characterised by a larger temporal extent of the cloud structure compared to ablation." * It would be interesting here to report the temporal extent of the cloud structure relative to the snowfall event duration because we would expect the persistence of cloud structures at least during the snowfall event. According to AWS, for SMB+ the temporal extent of the cloud structure is 252% of the snowfall event duration, while for SMB- it is 263%, so higher. Opposite trend for ERAI, with 277% for SMB+ and 243% for SMB-. So, absolutely specking, accumulation events are characterized by a larger temporal extent of the cloud structure, but relatively speaking, some other considerations could be done.**

This is a valid comment by the reviewer. The strong link between the temporal extent of the cloud structure and the duration of snowfall has not been considered. This has been clarified in the text and some conclusions are adapted.

*It is noted that relatively speaking, the temporal extent of the cloud structure equals on average 260 % of the duration of snowfall for both the accumulation and ablation events using both methods. It is therefore concluded that accumulation events are characterised by a larger temporal extent of the event compared to ablation, which is deduced from both the duration of snowfall and the temporal extent of the cloud structure.*

*In the previous section, synoptic events with snowfall lead to both accumulation and ablation, depending on the duration of the event (i.e. the duration of snowfall and temporal extent of the cloud structure).*

*The distinction between accumulation and ablation events during snowfall was correlated to the duration of the event (i.e. the duration of snowfall and the temporal cloud extent).*

**16. p.12 l.13-14: "From this, the ceilometer was able to detect 486 cm (i.e. the sum of all height changes during events for which blowing snow was detected)" * The ceilometer is not able to detect the snow height, it can detect blowing snow and then the acoustic height sensor can measure the height changes. Please re-word the sentence.**

This is a correct remark. The statement was adapted.

*In total, all 50 snowfall events attained for 542 cm of height changes (both accumulation and ablation; detected by the AWS). During the periods when the ceilometer recorded blowing snow, the AWS detected 486 cm of height changes (i.e. the sum of all height changes during events for which blowing snow was detected), showing the potential of the ceilometer to detect blowing snow during snowfall events.*

**17. p.12 l.16-17: "During all snowfall events, a total amount of 230 mm w.e. (approximately 230 cm in case fresh snow density equals 100 kg m$^{-3}$) was registered by the MRR, which is lower than the height changes recorded by the AWS (542 cm). This indicates the importance of the continuous movement of snow during snowfall events." * Without any information about snow density, the conversion of 1 mm w.e. to 1 cm cannot be considered realistic. PIP information should be used case by case to convert the w.e. to height and only after comparing the results to AWS measures. On the contrary, if in this work density information are used somehow for the conversion, it should be reported in the manuscript. Without clarifying this point, any conclusion made from this comparison cannot be considered reliable.**

This is a valid statement. Information about the density of snowfall at the PE station can be retrieved from the optical disdrometer (Precipitation Imaging Package) following the approach of Tiira et al. (2016). However, since we do not have precipitation gauge measurements at PE station, we are bound to the relation between particle diameter and snow density retrieved in Tiira et al. (2016). Over the PE station, the median diameter of the snow particles equals 0.7 mm. Using Eq. 14 of Tiira et al. (2016), a mean density of 323 kg m$^{-3}$ is obtained. Secondly, each year, the average density of the snow that has accumulated in the past year is calculated from snow pit measurements at the PE station. Here, yearly average snow densities varying between 309-395 kg m$^{-3}$ are obtained. The last method is preferred as these consist of measurements obtained at the PE station. Whenever a conversion between snow heights and water equivalent is needed, the yearly snowfall densities measured in the snow pit are used.

*During all snowfall events, a total amount of 230 mm w.e. was recorded. Based on yearly density measurements in the uppermost layers of the snow pack, yearly average values between 309-395 kg/m$^3$ are reported. This results in 58-75 cm accumulation at the surface if no other processes are at play. This is significantly lower than the total height changes recorded by the AWS during all snowfall events (542 cm).*

**18. p.15 l.14: "The ceilometer was only able to detect 274 cm" * Again, the ceilometer detects the blowing snow, not the snow height.**

Following comment 16, we also adapted this sentence.

*When the ceilometer detected blowing snow, only 274 cm of height changes were detected by*

*the AWS, indicating that the blowing snow layer only has a limited vertical extent and that the transport of snow is restricted to shallow layers close to the ground during these type of events.*

**19. p.16 l.5: "both accumulation as ablation" * "and" or "as well as".**

This sentence was removed from the original manuscript.

**20. p.16 l.7: "The distinction between accumulation and ablation events during snowfall was attributed to the duration of the event and the temporal cloud extent" * I would say "was related" or "we found a correlation between..." instead of "was attributed".**

The sentence has been adapted accordingly.

*The distinction between accumulation and ablation events during snowfall was correlated to the duration of the event (i.e. the duration of snowfall and the temporal cloud extent).*

**21. Supplement p.3 l.21: "influenced".**

The sentence has been adapted.

*In circulation type 1 (C1) the Antarctic circumpolar trough is clearly visible showing a high pressure bulge over Dronning Maud Land.*

**22. Fig.4a: "Wind rose showing the speed and direction of the snowfall events" * "Wind rose showing the speed and direction of the wind during snowfall events".**

The sentence has been adapted accordingly.

*Wind rose showing the hourly speed and direction of the wind recorded by the AWS during snowfall events detected by the MRR.*

**23. Fig. 5a: It is difficult to appreciate lines color differences, I would suggest to change the colorscale. Moreover, a legend describing the different lines would be useful.**

We have adapted the figure in order to improve readability. At first, a zoom to the Southern ocean north of Dronning Maud Land is made. The marker sizes of the line are enlarged in order to easier detect differences between the different lines. We choose to keep the colors as a sequential colorscheme is useful for denoting air parcels with different starting heights. We clarified the differences between the trajectories in the figure by stating that each line represents a different height and adapted the caption of the figure.

[revised manuscript text omitted]

---

## Author Comment (AC2) · 22 Mar 2018

**Response to Reviewer 2 Comments:**

**How does the ice sheet surface mass balance relate to snowfall? Insights from a ground-based precipitation radar in East Antarctica**

Niels Souverijns, Alexandra Gossart, Irina V. Gorodetskaya, Stef Lhermitte, Alexander Mangold, Quentin Laffineur, Andy Delcloo, Nicole P.M. van Lipzig

March 22, 2018

For clarifying our answers to the reviewers' comments, the following color scheme is used: comments of the reviewer are denoted in blue, our answers are denoted in black and quotes from the revised text are in green. Figure, section and table references refer to the revised manuscript. References including an "R" refer to respective tables and figures in the response letter.

**1 General comments**

*1. This paper presents a compilation of data from a set of instruments (Micro Rain Radar, ceilometer, Automatic Weather Station, among others) over a time period of 37 months, at the Princess Elisabeth (PE) Station, Antarctica. The authors analyse the different situations leading to accumulation and ablation at this site and conclude that "SMB records cannot be considered a good proxy for snowfall at the local scale". The authors compare the typical accumulation and ablation situations to results from a cluster analysis that describes the main weather situations at PE. Results suggest that a large part of the accumulation/ablation at the station takes place after the main precipitation events because "the fresh snow is easily picked up and transported in shallow drifting snow layers to more inland locations", even when wind speed is relatively low (< 7 ms$^{-1}$). This latter conclusion is intriguing because it suggests that snow drift may transport snow to more inland locations (here I understand upslope), which is clearly the opposite to present knowledge. Indeed, in Antarctica, it is assumed that snow is mainly drifted downslope due to the occurrence of katabatic winds and that significant amounts of snow are even drifted away from the ice cap to the ocean (Scarchilli et al., 2010; Palm et al., 2017). If the authors' conclusion is true, this process may have large consequences on our vision of snow transport and on accumulation processes in the coast to plateau transition zone. As a consequence, a clear demonstration of this finding is key in the present study, but in the present form, I am not convinced by their demonstration and several verifications are required. In particular, the authors should refer to results from Libois et al. (2014) who performed snow height measurements at Dome C, Antarctica, and demonstrated that the motion of sastrugi during low winds may*

*induce local snow accumulation or erosion at a decameter scale, leading to local snow height variations which seem to be very similar to the accumulation/erosion events presented in the present paper. Libois et al. (2014) suggest that "at each drift event, significant amounts of snow are deposited over approximately 20% of the total area only". This justifies that small amounts of deposited snow may move locally with low winds, and may accumulate in another location nearby, after the end of the precipitation event. If my suggestion is true, then the snow is only moved locally and not transported from (or to) remote areas.*

The reviewer is thanked for the thorough review of the manuscript and the different suggestions that are made, significantly improving the manuscript and increasing confidence in the conclusions.

We agree that the current analysis of snow erosion is not demonstrating the fact that snow is transported over large distances or to inland locations and that the results presented in Libois et al. (2014) could as well be valid for the PE station or, more generally, for the coast to plateau transition zone. In Libois et al. (2014), a combination of automatic and manual snow height observations at several locations over a limited domain were combined with the detailed snowpack model Crocus. Libois et al. (2014) points out that there is a large decameter-scale variability resulting from snow erosion, which can vary a lot on the hourly time scale, due to the formation and movement of sastrugi.

In the analysis of Libois et al. (2014), hourly snow height measurements are not postprocessed as their goal was to determine the movement of these sastrugi in detail. They identify sastrugi formation and decay at their measurement locations, leading to high peaks in snow height measurements (see their Figure 2). These peaks only last for one / a few hours (i.e. the time the sastrugi is present at the measurement location). In our study, the interest lies in snow height changes over larger spatial scales. In order to only detect height changes that attribute for long-term changes in the local surface mass balance and not the movement of these sastrugi, a running mean of 24 hours is applied on the hourly snow height measurements at our station. This is now discussed in the manuscript.

As our observational dataset is limited to snow height measurements at one location, we cannot identify the movement of sastrugi. In order to verify our original conclusions, results of the state-of-the-art high resolution (5.5 km grid) RACMO2.3 climate simulation are analysed (Lenaerts et al., 2014; van Wessem et al., 2014; Lenaerts et al., 2017), which is coupled to a snow model including drifting snow (Lenaerts and van den Broeke, 2012a). This simulation over Dronning Maud Land is forced at its lateral boundaries by atmospheric profiles of the ERA-Interim reanalysis and is free to evolve in its inner spatial domain.

From this RACMO2.3 simulation, moments during which accumulation and ablation events occurred without precipitation (SMB +, S 0 and SMB -, S 0 respectively) are extracted. The RACMO2.3 simulation is driven by ERA-Interim and adequately simulates climatological and meteorological conditions near the surface (e.g. Lenaerts et al., 2014). As such, we are confident that the observed events by the AWS and MRR are also simulated by the model. However, biases might still persist as the model is allowed to evolve freely. For both accumulation and ablation events, the average snow erosion pattern is calculated, together with the average wind field. A paragraph describing the application of the RACMO2.3 simulation has been added to both the Material and Methods section and the Results and Discussion section.

The results of the analysis of the RACMO2.3 high-resolution climate model impact the conclusions. The analysis indicates an upstream movements of snow, following the synoptic meteorological conditions. Most of the snow is transported over similar heights or slightly upslope (accumulation from the areas north of the mountain ridge towards the PE station) or downslope (ablation). As such, we have adapted the conclusions so upslope or more inland transport is rephrased in order to agree with the new insights.

[revised manuscript text omitted]

***2. This paper is well written and presents an interesting comparison between different sensors, but key information is lacking and new results (when compared to the previous publications) should be better highlighted: 1. The authors do not sufficiently present the methods, measurements and their uncertainty, and the reader has to refer to previous papers from the same team in order to find the information (Van Tricht et al., 2014; Gorodetskaya et al., 2013, 2015; Souverijns et al., 2017; Gossart et al., 2017). In particular, the sensor uncertainties are not presented, which impedes getting an accurate idea of the quality of the results => Please give more information on sensors and methods in the manuscript.***

We agree that the material and methods section is too concise and does not give a full overview of the instrumentation, its uncertainty, application and the processing of the data. Several sentences have been added including references to relevant publications.

[revised manuscript text omitted]

**3. In particular, estimates are made in mm w.e. but there is no information on snow density at site. What is occurring with density changes caused by snow aging?**

This is a correct statement by the reviewer. Each year, the average density of snow that has accumulated in the past year is calculated from snow pit measurements at the PE station. From this, yearly average snow densities varying between 309-395 kg m$^{-3}$ are obtained. These snow density measurements are also used to convert changes in snow height measured by the acoustic height sensor of the AWS into mm w.e.. We clarified this in the text.

Snow aging is not considered in the analysis. The authors have tried to implement this effect, however, a comparison between AWS measurements and the theoretical snow densification models of Reijmer et al. (2005); Vionnet et al. (2012) showed very weak correlations. Due to the continuous redistribution of snow over time, local densification of snow is difficult to assess at the PE station.

*Each year, the average density of the snow that has accumulated in the past year is calculated from snow pit measurements at the PE station. Using these average yearly densities (varying between 309-375 kg m$^{-3}$), snow height changes are converted to changes in the local SMB (water equivalent) (Gorodetskaya et al., 2013, 2015). Densification due to snow aging is not taken into account.*

**4. What is the final uncertainty in ERds estimates?**

$ER_{ds}$ represents the total snow erosion (both accumulation and ablation) by the wind over the year. $ER_{ds}$ is not measured directly by any of the instrumentation at the PE station and is calculated as a residual term: $ER_{ds} = S-SU_{ds}-SU_s-SMB$. As such, its uncertainty is based on the uncertainties of all input components (Gorodetskaya et al., 2015). Since the magnitude of the two sublimation terms is small over the PE station, the uncertainty on the $ER_{ds}$ component is mainly determined by the uncertainty of the snowfall estimate. The SMB uncertainty is reduced by applying the 24 hour running mean. We therefore estimate the uncertainty on $ER_{ds}$ is slightly higher than the uncertainty on snowfall rates (which was approximately 60 %).

*$ER_{ds}$ can be calculated as the residual term after inverting Eq. 1 (Gorodetskaya et al., 2015). Its uncertainty is therefore based on the uncertainty of all other components of the local SMB (Eq. 1; Gorodetskaya et al., 2015) and is mainly determined by the snowfall component at the PE station.*

**5. What is the representativeness of one acoustic gauge measurement at decameter or kilometer scale?**

In Libois et al. (2014), a combination of automatic and manual snow height observations at several locations over a limited domain were combined with the detailed snowpack model Crocus. They pointed out that there is a large decameter-scale variability resulting from snow erosion, which can vary a lot on the hourly time scale, due to the formation and movement of sastrugi. In order to erase part of the sastrugi movements and only detect height changes that attribute for long-term changes in the local surface mass balance, a running mean of 24 hours is applied on the hourly snow height measurements at the PE station. Furthermore, the spatial scale and pattern of snow erosion is analysed using the RACMO2.3 climate simulation. For a detailed description including the changes in the manuscript we refer to general comment 1.

**6. Is it physically justified to compare local acoustic gauge measurements with radar data that are integrating precipitation amounts at a kilometer scale?**

The radar deployed at the PE station is a vertically pointing radar and does not operate at the kilometer scale. However, since the lowest measurement bin of the MRR is located at 300 m a.g.l., high wind speeds can also horizontally displace falling snow particles before they reach the surface. (From Souverijns et al. (2017)): "In order to tackle this problem, the height correction of Wood (2011) is applied to the MRR data, by extrapolating the trend in the lowest MRR vertical levels towards the surface to account for horizontal displacement and sublimation

below the lowest measurement level. This results, on average, in a decrease in Z of 1.66 dBz between the lowest measurement level (at 300 m a.g.l.) and the surface." As such, we account for the temporal and spatial mismatch between AWS measurements and the MRR. We added a sentence to the text referring to this correction.

*A correction for the horizontal and vertical displacement of snowfall between the MRR data acquisition level and the surface has been applied (Wood, 2011; Souverijns et al., 2017).*

**7. Are there any differences between precipitation estimates from Gorodetskaya et al. (2015) (where the ceilometer and the MRR data are combined to analyse precipitation and clouds statistics) and those presented here => please clarify these points.**

The snowfall rates presented in this study are based on the results of Souverijns et al. (2017). In Gorodetskaya et al. (2015), little information about snow particle microphysics was available at the PE station. They applied nine Ze-SR relationships from literature, derived for dry (unrimed) snowfall for 34.6-35 GHz radar frequency. Using a Monte Carlo approach, a random Ze-SR relationship was selected for each precipitation event from the total of nine in order to calculate the total annual snowfall amount. The total yearly snowfall amount is then calculated as the mean of the Monte Carlo normal distribution (Gorodetskaya et al., 2015). Without additional information on the particle size distribution, this is the best option, but it has to be kept in mind that this method does not give a good idea of the true uncertainty (which in this case depends solely on the variability of the chosen relationships).

In January 2016, a Precipitation Imaging Package optical disdrometer was installed at the PE station. Using this instrument, information about the microphysical properties of the snow particles (i.e. the particle size distribution) became available at the PE station. As such, it was possible to derive a mean Ze-SR relation specifically for this location. Also, a better idea of the uncertainty on the snowfall numbers was obtained, as several sources of uncertainty are taken into account: measurement uncertainty by the disdrometer, shape uncertainty, parameter uncertainty and snow storm individual differences (Souverijns et al., 2017). We clarified the new approach in the text and stated the improvement compared to the results of Gorodetskaya et al. (2015).

*In order to obtain reliable estimates of snowfall rates and their uncertainty, an optical dis-drometer (Precipitation Imaging Package Newman et al., 2009) was installed at the PE station. This instrument is based on a high speed camera and is able to obtain detailed information about snowflake microphysics (i.e. the particle size distribution). A correction for the horizontal and vertical displacement of snowfall between the MRR data acquisition level and the surface has been applied (Wood, 2011; Souverijns et al., 2017). Using this information, a relation between radar reflectivity measured by the MRR and snowfall rates was obtained: $Ze=18SR^{1.1}$. Furthermore, a constrain on the uncertainty of the resulting snowfall rates was obtained [-59 % +60 %] (10th-90th percentile) (Souverijns et al., 2017), which is a considerable reduction compared to earlier snowfall rate estimates at the PE station that were retrieved without any information on the snow particle microphysical characteristics (Gorodetskaya et al., 2015).*

**8. Almost all the ideas and conclusions have been presented in Gorodetskaya et al. (2013, 2015). The main interests here are the extent of the dataset which allows to make a statistical analysis on a long time scale...**

The two papers referred to, entitled "Cloud and precipitation properties from ground-based

remote sensing instruments in East Antarctica" and "Meteorological regimes and accumulation patterns at Utsteinen, Dronning Maud Land, East Antarctica: Analysis of two contrasting years" described the SMB components but did not investigate in depth the drivers and mechanisms for changes in the different components. The studies from Gorodetskaya were the starting point of this paper where, based on a statistical long-term analysis of observations, an explanation for accumulation, ablation and snowfall events is given. In the abstract and the conclusion, more emphasis is put on these underlying mechanisms.

Abstract: *Snowfall originates mainly from moist and warm air advected from lower latitudes associated with cyclone activity. However, snowfall events are not always associated with accumulation. During 38 % of the observed snowfall cases, the freshly-fallen snow is ablated by the wind during the course of the event. Generally, snow storms of longer duration and larger spatial extent have a higher chance to attain for accumulation at the local scale, while shorter events usually attain for ablation (on average 17 and 12 hours respectively). A large part of the accumulation at the station takes place when preceding snowfall events were occurring in synoptic upstream areas. This fresh snow is easily picked up and transported in shallow drifting snow layers over tens of kilometers, even when wind speeds are relatively low ($< 7$ $ms^{-1}$). Ablation events are mainly related to katabatic winds originating from the Antarctic plateau and the mountain ranges in the south. These dry winds are able to remove snow and lead to a decrease in the local SMB. This work highlights that the local SMB is strongly influenced by synoptic upstream conditions.*

Conclusions: *The distinction between accumulation and ablation events during snowfall was correlated to the duration of the event (i.e. the duration of snowfall and the temporal cloud extent). Longer and larger events attain for bigger areas with fresh snow deposition, allowing snow from synoptically upstream areas to be transported towards the PE stations.*

Conclusions: *During most of these accumulation events, snowfall took place upstream of the PE station, ENE of the station and to the north of the Sør Rondane mountain ridge, within the last 24 hours. Winds easily pick up the freshly fallen snow from upstream areas and are capable of transporting it over tens of kilometers, leading to accumulation over large spatial areas potentially to higher elevations. Ablation events originate more often from southerly flows and also occur shortly after snowfall events at the station.*

**9. the statistical approach (cluster analysis) made to retrieve the main weather situation at the PE station. However, this cluster analysis is different from the one presented in previous studies and it leads to a different weather atlas. Please justify the differences. Does the choice of cluster have an impact on present conclusions?**

In this work a cluster analysis based on large-scale circulation (500 hPa geopotential fields) is executed for the Dronning Maud Land region. Large-scale drivers of meteorological events can be identified from this analysis including the movement of cyclones in the Antarctic circumpolar trough. A clear distinction between synoptic and katabatic influences is possible from this analysis. It was identified that cyclone activity attributed for most of the precipitation amounts at the PE station and generally at coastal sites over the AIS (Fig. 4). Furthermore, the intensity of the cyclone was strongly linked to the amount of precipitation (Fig. 5). For accumulation and ablation events without snowfall, large overlaps between circulation patterns were observed. Both events occur during synoptic and katabatic conditions. However, there is a slight tendency towards katabatic flow in case of ablation.

In the study of Gorodetskaya et al. (2013) the hierarchical cluster analysis was executed solely based on observations of the AWS at the PE station (based on longwave incoming radi-

[Figure]

Figure R1: Event classification based on the hierarchical cluster approach of Gorodetskaya et al. (2013). TRANS = transitional regime, COLD = cold katabatic regime, WARM = warm synoptic regime.

ation, wind speed, pressure, relative humidity and the inversion temperature). Three clusters were defined namely cold katabatic, transitional synoptic, and warm synoptic. For a full description of these three regimes, we refer to Gorodetskaya et al. (2013). As requested by the reviewer, this classification was applied on the full observational record of the AWS. For each of the events defined in Section 2.2, there was checked to which cluster they were assigned (Fig. R1).

Similar conclusions can be deduced from these results compared to our cluster analysis based on large-scale circulation. During snowfall events, the atmosphere is mainly influenced by the warm synoptic regime. This regime attributes for high wind speeds, cloudy conditions and high relative humidity and is probably related to cyclone activity. Accumulation and ablation events show a large overlap in the regimes, which was also found in our cluster analysis based on large-scale circulation. No clear assignment to the cold katabatic regime is found for ablation events and the main difference between accumulation and ablation events is detected in the increased occurrence of the transitional regime during ablation events. This is probably attributed due to the fact that most of the accumulation and ablation events occur shortly after snowfall events, which is often during the transitional regime.

Both clustering methods obtain similar results. However, there are several advantages to our cluster analysis defining large-scale circulation patterns. First, it has been shown in this study that there is an important link between large-scale synoptics and the conditions attributing for accumulation and ablation at the surface. The analysis of the large-scale circulation gives an overview of the different synoptic conditions. Second, the analysis in Gorodetskaya et al. (2013) is solely based on information measured by the AWS at the PE station, whereas the circulation analysis offers spatial information and thereby enhances insight in the advection processes. The variables used in Gorodetskaya et al. (2013) to define the clusters do not give information about the source of the events. Precipitation events over Antarctica are always related to high values of relative humidity and wind speeds, but this does not necessarily give an idea of the origin of the event, which can easily be derived from the cluster analysis based on large-scale circulation or the backtrajectory analysis.

*This cluster analysis method gives similar results to the local meteorological regimes defined by Gorodetskaya et al. (2013), but increases insight into the large-scale circulation patterns and source regions of air advection.*

**10. The main conclusion (Accumulation and ablation also occur during non-**

*snowfall conditions) is already presented in Gorodetskaya et al. (2015), but the new idea is that snow may be transported from low lying regions => Please refer to Libois et al. (2014) paper, and try to see whether their analysis may help in understanding the snow height changes observed during low winds at the PE station.*

The results of Libois et al. (2014) are discussed now in the paper. We refer to general comment 1 for a complete discussion. The abstract and general conclusions were adapted based on these new insights (see general comment 1). The sentence to which the reviewer is referring has been removed from the abstract.

Abstract: *A large part of the accumulation at the station takes place when preceding snowfall events were occurring in synoptic upstream areas. This fresh snow is easily picked up and transported in shallow drifting snow layers over tens of kilometers, even when wind speeds are relatively low ($< 7\ ms^{-1}$).*

Conclusions: *Wind-driven accumulation and ablation also occur without snowfall at the PE station. These accumulation events have a tendency to take place during similar types of circulation as snowfall events, however, they are characterised by lower wind speeds. During most of these accumulation events, snowfall took place upstream of the PE station, ENE of the station and to the north of the Sør Rondane mountain ridge, within the last 24 hours. Winds easily pick up the freshly fallen snow from upstream areas and are capable of transporting it over tens of kilometers, leading to accumulation over large spatial areas potentially to higher elevations.*

*11. Several conclusions rely on assumptions made on the blowing snow processes, but these processes are not discussed according to current knowledge based on drifting snow measurements (Li and Pomeroy, 1997; Nishimura and Nemoto, 2005; Nishimura and Ishimaru, 2012; Scarchilli et al., 2010; Libois et al., 2014; Trouvilliez et al., 2014; Barral et al., 2014; Amory et al., 2015, 2017; Das et al., 2013). In particular, these publications present important information on the typical threshold wind speeds for snow transport. They also already discuss the impact of snow density, of snow aging, and of the sintering temperature on the threshold wind speeds. Finally, they present key information on the link between the "mean slope in the wind direction" and erosion/deposition processes. This knowledge is important to assess whether large drifting snow fluxes may be observed with low winds for fresh snow. Please refer to these publications and discuss the results accordingly.*

It is agreed that a thorough discussion on the blowing snow processes is currently missing. The discussion of the $ER_{ds}$ term in the introduction has been expanded using the literature proposed by the reviewer. Furthermore, the results obtained in the paper are discussed according to these references.

[revised manuscript text omitted]

**12. The situation of the PE station is not discussed in the text, but the station is located at the lee of a rock crest. Are these results site specific? Is there any information on the distribution of accumulation at a kilometer scale around the station (with stakes and acoustic gauges). What is the snow distribution proposed by a regional scale model such as RACMO2 at that site?**

The information about the station's location has been expanded. Furthermore, information about the location of the AWSs was added. Both the old and the new AWS are located approximately 300 m east of the Utsteinen Ridge and 1 km north-northeast of the Utsteinen Nunatak. There are two main wind directions recorded by the AWS, south (katabatic winds originating from the glacier) and ENE (related to synoptic activity; bringing most of the precipitation) (see e.g. Gorodetskaya et al. (2013)). During easterly winds the AWS is located on the windward side of the Nunatak. Mass balance measurements in the vicinity of the station depict little accumulation on the windward side (i.e. location of the AWS) and relatively little ablation on the leeward side (Pattyn et al., 2010). SMB estimates at high-resolution (5.5 km) are available from the RACMO2.3 simulation that was used to determine snow erosion patterns. An overview map of the annual SMB for Dronning Maud Land can be found in Lenaerts et al. (2014) (their Figure 5a). No peculiar trends are depicted in the area around the PE station.

*The PE station is located in the escarpment zone of the East Antarctic plateau (71° 57' S, 23° 21' E; 1392 m above sea level), 173 km from the coast, in Dronning Maud Land, north of the Sør Rondane mountain chain on Utsteinen ridge (Fig. 1). Meteorological conditions at the station are influenced by both synoptic weather conditions and katabatic winds. A detailed description of the site can be found in Pattyn et al. (2010) and Gorodetskaya et al. (2013).*

*The AWS was installed in February 2009 and is located approximately 300 m east of the station (and the MRR) on the windward side of Utsteinen ridge (see Section 2.1) in a zone of little accumulation (Pattyn et al., 2010).*

**13. To summarize, the dataset is really interesting in particular for model validation and deserves to be published. However, in its present state, this paper is not a sufficiently new contribution compared to Gorodetskaya et al. (2013, 2015). Before publication in the Cryosphere, I suggest that the authors compare their results to the available outputs from the RACMO2 regional scale circulation model. This model includes snowdrift processes and outputs may allow "validating" their assumption. If the authors prefer the use of another model, of course, it would make sense. For instance, Stefaan Lhermite is co-author of the present paper and already published a paper using the wind transport submodel SnowTran-3D (Groot Zwaaftink et al., 2013; Gascoin et al., 2013) => Is it possible to use this model here in order to see whether the assumption made on snow transport during low winds is physically supported or not?**

The comments discussed here have been addressed in general comment 1 where the RACMO2.3 simulation is chosen to compare with our results. It must be taken into account that RACMO2.3 cannot be used as validation since model evaluation of snowdrift is extremely challenging (see general comment 1).

**14. As a conclusion, I suggest the authors to make major revisions on their paper. I propose:1) to describe the differences with Gorodetskaya et al. (2013, 2015) and to develop the paper around the interest of the new weather atlas**

*presented here, 2) to present more clearly the accuracy of sensors and of their estimates, and (accounting for the uncertainty of estimations) to critically discuss the variations of the surface level measured with the acoustic gauge, 3) to discuss the potential discrepancies between signals which are describing very different spatial scales, 4) to propose a comparison with a model including drifting snow processes (for instance RACMO2) allowing to give clear evidence on the occurrence of drifting snow events transporting snow from lower altitude areas OR 5) to consider that sastrugi may move locally when the wind is not very strong (Libois et al., 2014). According to their opinion (point 4 or 5), I propose that the authors say whether processes only impact local snow redistribution or impact the regional scale snow redistribution.*

Most of the concluding remarks have been addressed above.

1. The conclusions and abstract have been modified to focus more on new results in this study (see general comment 8). The justification of the new weather atlas is discussed in general comment 9. The new estimate of snowfall rates compared to Gorodetskaya et al. (2015) is discussed in general comment 7. The cluster analysis using the AWS data as in Gorodetskaya et al. (2013) has been applied to the entire time period and showed both the consistency of the results and new insights (i.e. snow amount per regime).

2. The estimates and the uncertainty of the different sensors has been expanded in the text. This is discussed in general comments 2, 4 & 7. A critical discussion of the snow height changes measured by the AWS is offered in general comments 1 & 5.

3. The different spatial scales that are used by the different sensors are discussed now in the text: general comments 1, 5, 6 & 13, including minor comment 26.

4. Our findings were compared with the RACMO2.3 climate simulations (see general comment 1).

5. It was concluded that regional snow redistribution is a large driver during most of the events (see general comment 1).

**2 Minor comments**

*1. Abstract line 8: please remove "an unprecedented"*

The sentence has been adapted accordingly.

*To achieve this, a unique collocated set of ground-based and in-situ remote sensing instrumentation (Micro Rain Radar, ceilometer, Automatic Weather Station, among others) was set up operating for a time period of 37 months.*

*2. Lines 9-11: The authors write: "However, snowfall events are much more common than accumulation events. During 38% of the observed snowfall cases, the freshly-fallen snow is ablated by the wind during the course of the event. Generally, snow storms of longer duration have a higher chance to attain for accumulation at the local scale, while shorter events usually attain for ablation" This conclusion is very similar to the following one given in Gorodetskaya et al. (2015): "Large accumulation events ($> 10mm$ w.e. $day^{-1}$) during the radar-measurement period of 26 months were always associated with snowfall, but at the*

*same time other snowfall events did not always lead to accumulation.". Please clarify which are the new conclusions in the present paper.*

We agree that this structure puts too much focus on the existing conclusion 'snowfall occurs more often than accumulation'. As such, we have restructured this part of the abstract so the focus is put on the new results from this paper, namely the statistical analysis of the drivers for the differences between accumulation and ablation during snowfall events, including a clearer quantification.

*However, snowfall events are not always associated with accumulation. During 38 % of the observed snowfall cases, the freshly-fallen snow is ablated by the wind during the course of the event. Generally, snow storms of longer duration and larger spatial extent have a higher chance to attain for accumulation at the local scale, while shorter events usually attain for ablation (on average 17 and 12 hours respectively).*

*3. Line 12: "As such, SMB records cannot be considered a good proxy for snowfall at the local scale." => Considering the decameter to kilometer scale variability of the SMB, or alternating of megadunes/wind glaze' areas for instance, this conclusion is trivial. Please reformulate.*

It is true this conclusion is trivial. We have deleted the sentence from the abstract. The link with decameter to kilometer scale variability is reformulated in the abstract, results and the conclusions.

Abstract: *This fresh snow is easily picked up and transported in shallow drifting snow layers over tens of kilometers, even when wind speeds are relatively low ($< 7\ ms^{-1}$).*
Results: for an overview, we refer to general comment 1.
Conclusions: *Winds easily pick up the freshly fallen snow from upstream areas and are capable of transporting it over tens of kilometers, leading to accumulation over large spatial areas potentially at higher elevations.*

*4. Lines 13-15: "when preceding snowfall events were occurring in upstream coastal areas. This fresh snow is easily picked up and transported in shallow drifting snow layers to more inland locations, even when wind speed is relatively low ($< 7\ ms^{-1}$)." => please refer to Libois et al. (2014) where a potential explanation of such variations without precipitation is proposed. If the authors do not concur with these explanations, please demonstrate clearly that the snow can be transported upslope.*

This sentence have been adapted conform the new conclusions. "Upstream coastal" has been changed to "upstream", as the snow erosion pattern obtained from the RACMO2.3 simulation and the wind rose (Fig. 4a) show that the fresh snow is transported towards the PE station from the ENE, thus originating north of the Sør Rondane mountain chain. Most of the snow is transported over similar heights or slightly upslope. Further, Libois et al. (2014) proposed that snow height variations are mainly caused by sastrugi movement, showing a large decameter-scale variability. This was mainly addressed in general comment 1, where the RACMO2.3 climate simulation was compared to our results.

*A large part of the accumulation at the station takes place when preceding snowfall events were occurring in synoptic upstream areas. This fresh snow is easily picked up and transported*

*in shallow drifting snow layers over tens of kilometers, even when wind speeds are relatively low ($< 7\,ms^{-1}$).*

*As shown by the RACMO2.3 simulation and ERA-Interim (compare Fig. 9c with Fig. 3 & 4b), most snowfall occurs upstream of the PE station, north of the Sør Rondane mountain ridge and ENE of the station during the 24 hours preceding the accumulation event at the PE station. This freshly fallen snow has a low density and is easily redistributed as the wind speed threshold for drifting snow erosion is low (Li and Pomeroy, 1997; Mahesh et al., 2003; Trouvilliez et al., 2014).*

**5. Introduction: Page 1 - Line 20 "an important regulator of the present and future global climate and water cycle" => I don't understand this sentence. The oceans are currently a regulator since they absorb 93% of the global warming, but what do the authors mean with the word "regulator" in the case of Antarctica?**

We agree with the reviewer that "regulator of the global climate" is an exaggeration. We therefore rephrased the sentence to focus on the impact on sea level and the global water cycle.

*The Antarctic Ice Sheet (AIS), being currently the largest ice body on earth, is an important regulator of present and future sea level and the global water cycle (Vaughan et al., 2013; Previdi and Polvani, 2016).*

**6. Page 1 – Line 22: what do you mean with coupled climate/surface models? Do you mean GCM? Coupled AOGCM? AOGCM coupled with dynamic changes in ice surfaces. The cited publications are referring to very different models and do not clearly indicate what kind of model the authors are referring to.**

This statement has been clarified.

*Climate models (potentially coupled to an ocean or ice sheet model) play an important role in understanding and quantifying the contribution of the AIS to sea level (rise) (Gregory and Huybrechts, 2006; Rignot et al., 2011; Ligtenberg et al., 2013; DeConto and Pollard, 2016).*

**7. Page 2-Line 5: please refer to papers from Grazioli et al. (2017a,b) because these papers also offer new information on precipitation measurements in Antarctica.**

The references have been added, including Souverijns et al. (2017).

*It has been noted by several authors that in order to fully understand the impact of the AIS on future sea level rise, information on the individual components of the present-day AIS SMB, including snowfall measurements, is indispensable (van Lipzig et al., 2002; Rignot et al., 2011; Shepherd et al., 2012; Agosta et al., 2013; Gorodetskaya et al., 2015; Lenaerts et al., 2016; Grazioli et al., 2017a,b; Souverijns et al., 2017).*

**8. Page 2, Line 8: the authors cite: Vaughan et al. (1999); Magand et al. (2007). However, the database from Vaughan et al. (1999) has been updated by Favier et al. (2013) and Wang et al. (2016).**

This is indeed correct. The references have been added. Furthermore, also Favier et al.

(2017) was added.

*As such, most studies rely on stake measurements registering only the total change in snow height (Vaughan et al., 1999; van Lipzig et al., 2004; Genthon et al., 2005; Magand et al., 2007; Eisen et al., 2008; Agosta et al., 2012; Favier et al., 2013; Wang et al., 2016; Favier et al., 2017).*

**9. Page 2 Line 9: "Gorodetskaya et al. (2015) were the first to quantify the different terms" => "Gorodetskaya et al. (2015) quantified the different terms"**

The sentence has been adapted accordingly.

*Gorodetskaya et al. (2015) quantified the different terms of the local SMB in a systematic way for the Princess Elisabeth (PE) station and determined snowfall from radar measurements.*

**10. Equation 1: I do not understand how "Sublimation of the drifting snow" and ERds may be separated? if a snow flake is sublimated while it is drifted, this means that it has been eroded first. Is the mass loss accounted twice in the calculation? i.e., is it accounted for (1) in sublimation and (2) in ERds? please explain this formulation.**

This is an interesting point noted by the reviewer. $ER_{ds}$ is based on the transport of snow towards or away from a point of observation. However in this component, when the snow is moving, there is no interaction with the environment (i.e. no mass loss or gain). The $SU_{ds}$ component takes into account the interaction between the snow particles and the environment when snow is suspended in the air. This term includes the mass loss when suspended snow sublimates in undersaturated air. In our study, both the $ER_{ds}$ and the $SU_{ds}$ term are not measured directly. The $ER_{ds}$ component is calculated as a residual term (see Eq. 1), while the $SU_{ds}$ term is based on the parametrisations of Bintanja and Reijmer (2001); Déry and Yau (2001) which mainly depend on relative humidity and wind speed. In reality, it is impossible to separate these two terms based on observations. One example to measure the $ER_{ds}$ component in situ would be using a drifting snow station or ice mass balance buoys as proposed by e.g. Nishimura and Nemoto (2005); Leonard et al. (2012); Nishimura and Ishimaru (2012); Trouvilliez et al. (2014); Amory et al. (2017).

**11. Page 2, line 18: stake measurements and ice cores => please refer to Thomas et al. (2017).**

The reference was added. Furthermore, also a reference to Favier et al. (2013); Wang et al. (2016) was added.

*Although previous studies proposed a variety of techniques to calculate the local SMB, most are confined to measuring the sum of all components (net height change at the surface) using stake measurements and ice cores (Vaughan et al., 1999; Rotschky et al., 2007; Favier et al., 2013; Wang et al., 2016; Thomas et al., 2017).*

**12. Page 2, line 19: "the separate measurement of any of the components of the local SMB is considered a difficult task" => please also refer to Eisen et al. (2008); Gorodetskaya et al. (2015); Amory et al. (2017); Grazioli et al. (2017b).**

The references were added.

*Generally, the separate measurement of any of the components of the local SMB is considered a difficult task (King and Turner, 1997; Vaughan et al., 1999; van Lipzig et al., 2002; van den Broeke et al., 2004; Eisen et al., 2008; Gorodetskaya et al., 2015; Amory et al., 2017; Grazioli et al., 2017a).*

**13. Page 2, line 31: "Wind-induced accumulation / ablation by drifting / blowing snow over the AIS has an important impact on the local SMB" => please refer to Palm et al. (2017).**

The reference has been added, including Scarchilli et al. (2010).

*Wind-induced accumulation and ablation by blowing or drifting snow over the AIS has an important impact on the local SMB (Bromwich et al., 2004; Scarchilli et al., 2010; Palm et al., 2017).*

**14. Page 2, line 34: "using a network of snowdrift instrumentation (Leonard et al., 2012)" => please also refer to Nishimura and Nemoto (2005); Nishimura and Ishimaru (2012); Trouvilliez et al. (2014); Amory et al. (2017).**

The references have been added.

*The $ER_{ds}$ component can however only be measured accurately using a network of snowdrift instrumentation (Nishimura and Nemoto, 2005; Leonard et al., 2012; Nishimura and Ishimaru, 2012; Barral et al., 2014; Trouvilliez et al., 2014; Amory et al., 2017).*

**15. Page 2, line 35: "Neglecting this term might however lead to significant errors" => How much?**

Values from Lenaerts and van den Broeke (2012a) (based on their Figure 12) show that drifting snow can locally attribute to a removal of 40 % of the accumulated snowfall.

In the revised version of the manuscript, this sentence has been deleted.

**16. Page 3, line 10: "The total SMB or snow height can be measured by an Automatic Weather Station (AWS)," => this sentence is not correct because AWS do not offer any information on snow density. More generally there is no information on snow density in the present paper. Do the authors consider the same value of density all the time? How do the authors consider variations in density related to snow aging? Please reformulate and clearly describe how the snow density is considered in calculations.**

We agree this concept has not been explained well. Each year, the average density of snow that has accumulated in the past year is calculated from snow pit measurements at the PE station. From this, yearly average snow densities varying between 309-395 kg m$^{-3}$ are obtained. These snow density measurements are also used to convert changes in snow height measured by the acoustic height sensor of the AWS into mm w.e.. We clarified this in the text.

Snow aging is not considered in the analysis. The authors have tried to implement this effect, however, a comparison between AWS measurements and the theoretical snow densification models of Reijmer et al. (2005); Vionnet et al. (2012) showed very weak correlations. Due to the continuous redistribution of snow over time, local densification of snow is difficult to assess at the PE station.

*Each year, the average density of the snow that has accumulated in the past year is calculated from snow pit measurements at the PE station. Using these average yearly densities (varying between 309-375 kg m$^{-3}$), snow height changes are converted to changes in the local SMB (water equivalent) (Gorodetskaya et al., 2013, 2015). Densification due to snow aging is not taken into account.*

**17. Page 3, line 19: "Gorodetskaya et al. (2015) noted that snowfall events at the PE station do not necessarily contribute to accumulation or an increase in the height of the local SMB." => The authors write that their main conclusions have already been given in previous publications. Please describe what the new insights are? For instance, the present study may offer a more robust estimation of the frequency of different events.**

This comment has been addressed in general comment 8.

**18. Section 2.1: page 3, line 28: "In order to gain insight in the relation between snowfall and the SMB, reliable, high-frequency and long-term in situ observations are indispensable." => repetition with the introduction. this sentence may be removed.**

The sentence has been removed.

**19. Figure 1: the P-E station appears to be close to a blue ice area (located southward) and in the lee of a mountain ridge. Is there any impact of the location on local accumulation? For instance, when the wind is coming from the south, it is coming from the blue ice area and the ridge, which are characterized by erosion: snow is not available for transport, but these areas are in the vicinity of the PE station. Conversely, the areas located close to the PE station but in the ENE direction are covered by snow. Erosion and transport from these areas thus occur more easily, attaining for deposition at the PE station. This may suggest that snow is transported only over small spatial scales when the wind speed is low and not necessarily from further low-lying areas, in agreement with the processes described by Libois et al. (2014). As a consequence, the impact on accumulation distribution may be only local. What is the authors' opinion?**

The comment of the reviewer is indeed correct. South of the PE station, there are several blue ice zones visible at distances between 10-50 kilometers. The winds originating from the south (i.e. katabatic winds) are able to erode the surface at the PE station. For accumulation events, the regions ENE of the station and north of the Sør Rondane mountains often experienced snowfall accumulation in the past 24 hours. As such, a lot of fresh snow is available to easily be transported towards the PE station in case the wind stays stable after the snowfall event. This has been confirmed by the RACMO2.3 simulation results (see general comment 1 for a thorough discussion) and shows that the transport of snow can take place over tens of kilometers.

*During calm conditions, a katabatic flow manifests originating from the mountain ridge and the Antarctic plateau (Parish and Cassano, 2003). These areas south of the PE station received generally less snowfall during the preceding snowfall event compared to the areas north of the Sør Rondane mountains (Fig. 9c & Palerme et al. (2014)). They are therefore capable of picking up fresh snow at the PE station attaining for ablation (Scarchilli et al., 2010). Similar conclusions can be drawn from the analysis of the RACMO2.3 climate model output, showing large areas of snow removal at the edge of the mountain ridge (Fig. 9b) and was also identified in RACMO2.3 by Lenaerts and van den Broeke (2012b).*

*To assess the large-scale spatial patterns of snowdrift transport, our observed record is compared with output from a regional climate model. Here, we use the high-resolution RACMO2.3 climate model output. In this simulation, a number of 36 modelled accumulation events attributed for a net snow accumulation of more than 1 kg $m^{-2}$ at the pixel corresponding to the PE station location in the RACMO2.3 simulation (which equals to height changes between 0.26-0.33 cm assuming snow densities between 309-375 kg $m^{-3}$), while 116 modelled ablation events attained for net snowfall removal of more than 1 kg $m^{-2}$. This is approximately 35 % of the observed quantities by the AWS. There are several potential underlying causes for this difference: either, part of the snow erosion that is detected by the AWS occurs on the subgrid scale, or RACMO2.3 underestimates the blowing snow amount or its divergence. In fact, Lenaerts et al. (2014) showed RACMO2.3 underestimates blowing snow amounts around the PE station. Generally, RCMs tend to underestimate blowing snow amounts over the whole of Antarctica (Amory et al., 2015). Almost all areas north of the mountain ridge attain for snow accumulation in the RACMO2.3 simulation when there is accumulation but no snowfall at the PE station (Fig. 9a). A fraction of 43 % of accumulation events occurs within 24 hours after a snowfall event at the PE station (i.e. 31 out of 72 events). As shown by the RACMO2.3 simulation and ERA-Interim (compare Fig. 9c with Fig. 3 & 4b), most snowfall occurs upstream of the PE station, north of the Sør Rondane mountain ridge and ENE of the station during the 24 hours preceding the accumulation event at the PE station. This freshly fallen snow has a low density and is easily redistributed as the wind speed threshold for drifting snow erosion is low (Li and Pomeroy, 1997; Mahesh et al., 2003; Trouvilliez et al., 2014). As such, in case the wind pattern stays stable from the ENE after the snowfall, it can be transported upstream attaining for accumulation at more westerly locations, such as the PE station.*

**20. Page 4, line 13: "snowfall rates was obtained including a constrain on its uncertainty (Souverijns et al., 2017)." => please give uncertainty values for all the sensors.**

This comment was addressed in general comment 2 where the uncertainty values for all the instruments and sensors are discussed.

**21. Page 4, line 16: "Furthermore, it is equipped by an acoustic height sensor, which is able to measure snow height changes with an accuracy of 1 cm on an hourly time resolution." and later, "resolution. Snowfall events are defined as exceeding the threshold of 1 mm w.e. during the continuous duration of snowfall measured by the MRR (this corresponds to 1 cm of snow when the density of fresh snow equals 100 kg $m^{-3}$)." => (1) This threshold may be too high to depict feeble precipitation: do results depend on this threshold? => (2) In Libois et al. (2014), they write: "we chose to set the density of fresh snow to 170 kg $m^{-3}$, which corresponds to the fifth lowest percentile of the measured surface densities at Dome C during the 2012–2013 campaign.". In this study at Dome**

*C, where diamond dust and surface hoar may occur, the fresh snow density is already 70% higher than at PE. Please justify the choice of this very low snow density value (100 kg m$^{-3}$). Is there any impact of this choice on the comparison between MRR precipitation and surface level variations? If higher density values are considered, then the 1cm threshold would lead to neglect events with higher accumulation?*

The conversion of snowfall amounts to height changes is indeed inappropriate in this paragraph, has no added value in this part of the text as it is not used further. It has been removed from this section.

Information about the density of snowfall at the PE station can be retrieved from the optical disdrometer (Precipitation Imaging Package) following the approach of Tiira et al. (2016). However, since we do not have precipitation gauge measurements at PE station, we are bound to the relation between particle diameter and snow density retrieved in Tiira et al. (2016). Over the PE station, the median diameter of the snow particles equals 0.7 mm. Using Eq. 14 of Tiira et al. (2016), a mean density of 323 kg m$^{-3}$ is obtained. Secondly, each year, the average density of the snow that has accumulated in the past year is calculated from snow pit measurements at the PE station. Here, yearly average snow densities varying between 309-395 kg m$^{-3}$ are obtained. The last method is preferred as these consist of measurements obtained at the PE station. Whenever a conversion between snow heights and water equivalent is needed, the yearly snowfall densities measured in the snow pit are used.

*During all snowfall events, a total amount of 230 mm w.e. was recorded. Based on yearly density measurements in the uppermost layers of the snow pack, yearly average values between 309-395 kg/m$^3$ are reported.*

*22. page 5, line 4: "Recently, an algorithm for the detection of blowing snow by the use of a ceilometer was developed" => in Gorodetskaya et al. (2015), a combined analysis of the ceilometer and of the MRR is done to get information on clouds and precipitation, here the ceilometer is used to assess blowing snow events? Here, do the authors use the ceilometer to also analyse the clouds characteristics?*

The ceilometer is a very good tool to analyse cloud properties over the station. For the analysis of the temporal cloud extent, it would be a logical choice. However, the ceilometer had several inactive periods during which both the MRR and the AWS were measuring. Depending on the type of event, the availability of the ceilometer varies between 55 % and 74 % (Table R1). As such, not enough events are available to determine reliable statistics on cloud extent. In that respect, we used longwave downward radiation as a proxy for cloud cover from the AWS. The advantage of variables retrieved by the AWS is the fact that it is battery-powered and therefore not affected by power-cuts at the station. As such, a long-term record of meteorological data, including information about radiation, is available (see also Fig. S1 in the Supplements).
A sentence about this problem was added to the text.

*Ceilometer cloud observations are not used to determine the temporal extent of the cloud system, as the collocated time period of corresponding measurements is insufficient.*

*23. page 5, line 8: "The minimal height of the blowing snow layer to be detected by the ceilometer equals 30 m at the PE station." => please give sensor*

| | Total number of events | Events with ceilometer operational | % |
|---|---|---|---|
| SMB +, S + | 31 | 20 | 65 % |
| SMB -, S + | 19 | 14 | 74 % |
| SMB +, S 0 | 72 | 49 | 68 % |
| SMB -, S 0 | 87 | 48 | 55 % |

Table R1: Overview of the number of events per type and the amount of events for which ceilometer data is available.

**uncertainty and the impact on the blowing snow flux estimate.**

Information about the ceilometer's ability to capture blowing snow has been added. It is also noted that the ceilometer is not able to determine the blowing snow flux. We therefore limit ourselves to the determination of the blowing snow frequency. The ability of the instrument to detect the frequency is compared to visual observations.

*The minimal height of the blowing snow layer to be detected by the ceilometer equals 30 m at the PE station. The temporal resolution equals 15 sec, while the vertical height resolution is 10 m. The quantification of blowing snow particle density, shape or number from the ceilometer attenuated backscatter signal is very uncertain (Wiegner et al., 2014). As such, this study is limited to the determination of the blowing snow frequency. Results show that 78 % of the detected events by the ceilometer are in agreement with visual observations at Neumayer III station and that blowing snow occurrence strongly depends on fresh snow availability in addition to wind speed (Gossart et al., 2017).*

*24. page 5, line 9: "Surface and snowdrift sublimation are quantified using the approach of Thiery et al. (2012)" => please give uncertainty values.*

The quantification and uncertainty on the sublimation terms has been expanded and is discussed in general comment 2.

*25. page 5, line 18: "which is unprecedented for the Antarctic region." => There are many other similar datasets in Antarctica available for many years but with different sensors and focus (e.g. the AWS network). Please remove or reformulate.*

We agree that networks of AWS, SMB measurements, among others exist over the AIS. However, we stress the presence of all the different instrumentation at one location. As far as we know, the PE station is the only station that measures precipitation, the SMB components and meteorology at the same location. Efforts are currently made at the Mario Zucchelli and Dumont D'Urville station, with the application of a Micro Rain Radar, but no closure of the SMB is achieved yet. We rephrased the sentence to stress this.

*In the end, 37 months of collocated precipitation, total SMB and meteorological data are available at the same location, which is unprecedented for the Antarctic region.*

*26. Page 5, line 22: "Nevertheless, since the AWS measures the total SMB directly, ERds can be calculated as the residual term after inverting Eq. 1" => this calculation indicates that ERds values include the sum of all other uncertainties. What is the accuracy of this term? Moreover, the different sensors used here*

*refer to different spatial scales (km$^2$ scale for the MRR and ceilometer, but a few m$^2$ for the ultrasonic gauges). Is it physically justified to directly compare these very different scales? In order to accurately compare the MRR/ceilometer data with surface height data, it may be necessary to consider snow height variations given by multiple sensors distributed over the area. In particular, referring to Libois et al. (2014) publication, is it possible that snow accumulates in depressions or along small snow barchans, which may move due to snow reptation? For instance, please observe the 3 large increases and decreases of the surface level in January, February and March 2012 (Figure 2).*

The accuracy and the uncertainty of the $ER_{ds}$ term are discussed in general comment 4. A discussion about the spatial scale of the MRR measurements can be found in general comment 6 and is also applicable on the ceilometer.

In Figure 2 of the main paper, three peaks in local SMB are visible in January, February and March 2012. As the reviewer suggests, these events might be related to the movement of small snow barchans. As the movement of these barchans attributes for short increases or decreases in snow height, most of them are erased by the application of a 24 hour running mean on the AWS data (see general comment 1). We acknowledge that these reptations might impact the results of our analysis and therefore added an extra condition in order to include accumulation and ablation events to our analysis. In case an increase / decrease in snow height is followed by a strong decrease / increase due to which 80 % of the original height change is removed within 48 hours, the events are discarded. This was for example the case for the three events the reviewer detected in Figure 2.

*Apart from this, temporary peaks (i.e. with a strong decay within 48 hours) in the snow height records are excluded from the analysis.*

*27. Section 2.3 Page 6, Line 12: give more details on the method, and particularly on measurement uncertainty.*

Extra information about the meteorological and snowfall data acquisition, processing and their uncertainty is given in Section 2.1. This comment has been discussed in general comment 2.

*28. Page 6, line 30: "The SANDRA optimisation algorithm was selected to perform this cluster analysis," => Please explain the interest of this new cluster analysis. Why is it more interesting and robust than the one used in Gorodetskaya et al. (2013)? The weather atlas presented in the two different papers looks different, please justify these differences.*

Some explanations on the SANDRA cluster algorithm was added including some information on the selection of this algorithm. An elaborate discussion can be found in general comment 2. An extensive comparison between the cluster analysis performed here and the one in Gorodetskaya et al. (2013) is presented in general comment 9. The advantages of our method compared to Gorodetskaya et al. (2013) for the interpretation of the results is also discussed.

*29. Page 7, lines 3-6: "a total of nine circulation patterns was selected. [...] during the four types of events attaining for a change in the local SMB" => in figures, 6 patterns are presented. Please clarify. Which are the four main patterns within the 6 patterns?*

This is indeed a typo in the manuscript. As depicted in the figure, there are only six patterns instead of nine. The text has been adapted.

The four types of events that are stated denote the four types between which we differentiate i.e.

- SMB +, S +

- SMB -, S +

- SMB +, S 0

- SMB -, S 0

This is explained more clearly in the text now with a reference to the corresponding section.

*Based on this climatology, the dominant circulation present during the four types of events attaining for a change in the local SMB defined in Sect. 2.2 are determined.*

**30. Page 7, line 13: "are converted to water equivalent values" => please indicate how density changes are considered in the first snow layers?**

This remark has been discussed in general comment 3. Furthermore, we adapted the sentence denoted by the reviewer.

*The four components of the local SMB, snowfall, surface sublimation, drifting snow sublimation and wind-induced accumulation / ablation, are converted to water equivalent values (see Sect. 1; Gorodetskaya et al., 2013, 2015).*

**31. Page 8 Figure 2: This figure is similar to Figure 9 from Gorodetskaya et al. (2015). I suggest the authors to present their results over the whole 37 months.**

It is acknowledged that Fig. 2 resembles Fig. 9 of Gorodetskaya et al. (2015). The main difference between both is the way of visualisation. Snowfall is depicted as the positive term of the SMB, while the sublimation terms are more clearly visible as ablation terms. Furthermore, the snowfall product is now updated based on the results of Souverijns et al. (2017) leading to less uncertain values compared to Gorodetskaya et al. (2015). This figure is depicted in this paper as it gives a good overview and visualisation of the behaviour of the different components of the SMB. Different ablation and accumulation events can easily be detected and also an idea of the magnitude and frequency of occurrence of different events and components of the SMB is visible. An attempt has been made to visualise the full time period. However, due to the big data gaps (i.e. all boreal winters except 2012; see Fig. S1 in the Supplement) no satisfactory result is obtained. For example, during the boreal winter of 2013 a huge accumulation event took place recorded by the AWS. However, since the MRR was not operational, one could not check if it was related to snowfall or snow drift. When one tries to visualise the 37 months leaving all data gaps out, a similar problem arises. Every data gap leads to large jumps in the SMB and snowfall record, which would lead to wrong or difficult interpretations by readers of the paper according to the authors. As 2012 still remains the only year with continuous data from both the MRR and the AWS, it is chosen to only visualise this year.

[Figure]

Figure R2: Time series comparison between daily precipitation amounts recorded by the MRR and the pixel nearest to the PE station in the ERA-Interim dataset.

**32. Page 8 Line 5: did the authors compare Precipitation estimates from ERA-Interim with precipitation from the MRR? is it possible to consider that interpretation of the cluster and backtrajectory analyses are robust If ERA precipitations do not fit with field measurements?**

This is an important concern raised by the reviewer. Both the cluster and backtrajectory analyses are based on ERA-Interim large-scale atmospheric flow, which in its turn influences precipitation numbers over the PE station (i.e. the circumpolar trough). The link between events recorded by instrumentation at the PE station and the results of the cluster and backtrajectory analysis is made several times. Therefore, it is necessary to check if ERA-Interim is able to simulate meteorological events at the PE station. For this, we analyse precipitation events measured both by ERA-Interim and the MRR.

First, a time series comparison was executed between MRR precipitation numbers and the precipitation total in the ERA-Interim pixel closest to the PE station. Over the full operational period of 37 months, the MRR measured a total snowfall amount of 1208 mm w.e., while ERA-Interim recorded 946 mm w.e. in the pixel over the PE station. An example of snowfall numbers recorded by both instruments for one month (representative for the whole time series) is given in Fig. R2. In this month several snowfall events are detected. From this, two were related to cyclone activity ($\pm$ 8th of February and $\pm$ 19th of February). These events are well-captured in both ERA-Interim and the MRR, although the time span of the event is larger in ERA-Interim. This might be related to the size of the grid box in ERA-Interim, which is 0.75° x 0.75°. This is visible in all precipitation events recorded by both products. Some events are visible in the ERA-Interim dataset that are not detected by the MRR. These events are however often small and are usually not driven by large-scale circulation.

In order to confirm that precipitation events are captured by both the MRR and the ERA-Interim reanalysis, a scatter plot comparison is executed (Fig. R3). Most of the large precipitation events are recorded by both instruments, confirming the results depicted in Fig. R2. There is however a constant underestimation of these larger precipitation events by ERA-Interim, probably related to the large pixel size, smoothing out the precipitation signal. Apart from this, the Pearson correlation coefficient between both products is calculated (excluding zero precipitation amounts measured by both instruments simultaneously), equalling 0.89. These

[Figure]

Figure R3: Comparison of daily precipitation numbers of the ERA-Interim reanalysis and precipitation rates recorded by the MRR.

results show high confidence in the precipitation comparison between ERA-Interim and the MRR, validating the usage of ERA-Interim as a product to derive large-scale circulation. However, it is noted that ERA-Interim is a potential source of error. This has been mentioned in the manuscript.

*Precipitation estimates are obtained from ERA-Interim, currently considered the best Antarctic-wide precipitation product, however still strongly biased (Bromwich et al., 2011). Misrepresentations of large-scale atmospheric flow and precipitation might therefore impact the results.*

**33. Section 3.3 Page 8, line 10: "in total 50 independent snowfall episodes were detected attaining for at least 1 mm w.e. at the PE station" => Libois et al. (2014) present a statistical analysis and write that: "Under the ergodic hypothesis, we conclude that at each drift event, significant amounts of snow are deposited over approximately 20% of the total area only". This suggests that there is only a low probability to capture snow accumulation during drifting snow events with only one acoustic gauge. Do the authors believe that their results may change if they had access to the mean variations given by 3 or 5 sonic gauges separated by 10m from each other?**

This comment has been discussed in general comment 1.

**34. Page 9: Figure 3. Cases 3 and 4 look very similar. Please indicate more clearly in the text and in captions which are the Cases C1, C2, etc... This interpretation of the cluster should be strengthened in the text and compared with previous papers. Do the conclusions depend on the clusters analysis?**

A basic interpretation of the cluster analysis and the individual circulation patterns has been added to the main text. The importance of showing the variability between circulation types 2-4 is indicated as it shows the typical cyclone movement in the circumpolar trough. A more extensive description of each circulation pattern can be retrieved from the Supplement. The caption of Fig. 3 has also been expanded. An extensive comparison between the cluster analysis performed here and the one in Gorodetskaya et al. (2013) is presented in general comment 9. The advantages of our method compared to Gorodetskaya et al. (2013) for the interpretation of the results is also discussed.

*The sequence of circulation patterns depicted in Fig. 3 is typical for the Antarctic region (Simmonds et al., 2003). In circulation type 1 (C1) the Antarctic circumpolar trough is clearly visible showing a high pressure bulge over Dronning Maud Land. East and west, we can see two low pressure cells. In C2-C5 the typical movement of a low pressure cyclone from the west to the east is depicted, largely influencing meteorological conditions at the surface. Apart from the circulation, also the average precipitation amounts associated to each circulation pattern are shown (Fig. 3). Precipitation estimates are obtained from ERA-Interim, currently considered the best Antarctic-wide precipitation product, however still strongly biased (Bromwich et al., 2011). Misrepresentations of large-scale atmospheric flow and precipitation might therefore impact the results. A strong link between the location of the cyclone and precipitation amounts is present. The cyclone is capable of transporting marine air towards the AIS. These marine air masses have the potential to take up moisture, potentially attaining for precipitation at the continent. A detailed description of each circulation pattern individually can be retrieved from the Supplement.*

*Weather atlas illustrating the circulation climatology over Dronning Maud Land. Thick lines denote the 500 hPa geopotential fields, while blue colours show average precipitation amounts linked to this circulation. The red dot indicates the location of the Princess Elisabeth station, while the green dot denotes the location over the ocean used for calculating the pressure gradient (Section 3.3). The Antarctic circumpolar trough is identified. The depiction sequence of the circulation types describes the typical west-east movement of the cyclones.*

**35. Page 11, line 12: "From the total number of snowfall events, 31 resulted in accumulation (62%), while 19 led to ablation (38%)." => is there any relationship with precipitation intensities or amounts given by the MRR data and accumulation/ablation occurrences?**

Average precipitation numbers for snowfall events attaining for accumulation and ablation have been calculated. Accumulation events attain on average for 5.3 mm w.e., while for ablation events this is 3.6 mm w.e.. Accumulation events attain for higher numbers of precipitation. This can partly be attributed to the larger size / duration of the precipitation event. Higher precipitation numbers also increase the chance for accumulation at the local scale (depending on the synoptic situation). We have added the average precipitation numbers for these events to the text.

*From the total number of snowfall events, 31 resulted in accumulation (62 %, attaining on average for 5.3 mm w.e. of snowfall), while 19 led to ablation (38 %, attaining on average for 3.6 mm w.e. of snowfall; Table 1).*

**36. Page 12, line 5: please discuss this paragraph considering conclusions from Libois et al. (2014).**

This comment has been addressed in general comment 1 & 11. In this paragraph we refer to the next section, where the link to the conclusions from Libois et al. (2014) is made.

*In order to validate this hypothesis, information about the scale of snow displacement is*

*necessary (see Section 3.4).*

**37. Page 12, line 16: "During all snowfall events, a total amount of 230 mm w.e. (approximately 230 cm in case fresh snow density equals 100 kg m$^{-3}$) was registered by the MRR, which is lower than the height changes recorded by the AWS (542 cm)." => what is the accuracy of both sensors? please also discuss the impact of snow density value on results.**

The accuracy of snowfall rates deduced from the MRR and the disdrometer has been discussed in general comment 2. The accuracy of the acoustic height sensor and the MRR have been discussed in general comment 5. The density of fresh snow has been discussed in general comment 3.

**38. Is it possible to consider that the AWS is located in a small overaccumulation zone (due to gravity waves at the lee of the mountain crest for instance)? I propose the authors to discuss these differences.**

The PE station is built on Utsteinen Ridge at the northern foot of the Sør Rondane Mountain Range (see e.g. Pattyn et al. (2010); Gorodetskaya et al. (2013). Both the old and the new AWS are located approximately 300 m east of the Utsteinen Ridge and 1 km north-northeast of the Utsteinen Nunatak. There are two main wind directions recorded by the AWS, south (katabatic winds originating from the glacier) and east (related to synoptic activity; bringing most of the precipitation) (see e.g. Gorodetskaya et al. (2013)). During easterly winds the AWS is located on the windward side of the Nunatak. Mass balance measurements in the vicinity of the station depict small accumulation to the east (i.e. location of the AWS) and relatively little ablation to the west of Utsteinen Ridge (Pattyn et al., 2010). As such, the AWS is not influenced largely by the local topography. We also refer to minor comment 12.

*The AWS was installed in February 2009 and is located approximately 300 m east of the station (and the MRR) on the windward side of Utsteinen ridge (see Section 2.1) in a zone of little accumulation (Pattyn et al., 2010).*

**39. Page 13, line 1: "Accumulation records are therefore not advised to be used as a proxy for snowfall over East Antarctica." => please, replace "accumulation" by "punctual accumulation". Indeed, I suppose that the conclusion would be different if the authors had access to many stake/ice cores/sonic gauge data over a large area.**

Based on the analyses executed above regarding the RACMO2.3 simulation (see e.g. general comment 1), not only punctual accumulation is a bad proxy for snowfall, but also accumulation over larger spatial areas. As such, we have added punctual between brackets.

*(Punctual) accumulation records are therefore not advised to be used as a proxy for snowfall over East Antarctica.*

**40. Section 3.4 Page 13, line 6: "snow displacement events also occur without the presence of snowfall." => what is horizontal scale here? Deca-centimeter? Kilometer? Decakilometer?**

This comment has been addressed in general comment 1.

*41. Page 14, Line 2: "limiting the regions that receive snowfall to coastal areas" => how do the authors observe that precipitation occur in coastal areas? Do they use ERA-interim outputs?*

Snowfall numbers are generally higher at coastal sites compared to inland regions. The precipitation distribution over the AIS is largely defined by orography. This is observed in several precipitation products (e.g. Palerme et al., 2014; Behrangi et al., 2016). Apart from the climatological time scale, this has also been observed for individual events based on ERA-Interim data (Gorodetskaya et al., 2014).

*42. Page 14, line3: "to be limited to the coastal areas, not reaching the PE station" => I suppose that the MRR data are used in order to reach this conclusion. However, Gorodetskaya et al. (2015) write: "While MRR misses the most feeble precipitation (virga or snowfall) with Ze < -8 dBz, its sensitivity is sufficient to detect typical precipitation at the site. [...] and ice clouds or weak precipitation not detected by MRR (22% of the total period or 63% of the overcast)." => I suppose that feeble precipitations are not always detected by the MRR. If it were the case, the MRR would not be sufficient to demonstrate that no precipitation occurred. Please clarify.*

Since the publication of Gorodetskaya et al. (2015), the processing method of Maahn and Kollias (2012) was updated in order to fully exploit the MRR hardware in case of solid precipitation, increasing its sensitivity up to -14 and -8 dBz, depending on vertical range (Souverijns et al., 2017). At the heights closest to the surface, the sensitivity equals -14 dBz. Using the Ze-SR relation obtained by Souverijns et al. (2017), this complies to a snowfall rate of approximately 0.004 mm w.e./h. In Gorodetskaya et al. (2015), the minimum detection limit equaled 0.014 mm w.e./h. With this new threshold, we are confident to capture all precipitation events occurring at the PE station.

*43. Page 14, line 16: "A fraction of 46% of the ablation events take place within 24 hours after a snowfall event. As stated in Sect. 3.3, snowfall events are mainly characterised by a ENE flow (Fig. 4a)." => do the authors estimate the "reptation" velocity (e.g. Nishimura et al., 2014) in order to estimate the distance over which the snow particles were transported until reaching PE station?*

An analysis of the reptation velocity would be out of the scope of the study. This analysis would need information about the blowing snow particle diameters, which is unavailable with the instrumentation at the PE station. Last year a particle counter was installed by a Swiss team in order to gain more information about the snowdrift particle characteristics. However, no concurrent measurements of this instrumentation with the MRR and AWS is currently available. Hopefully, in future, this data can be used to asses the reptation velocity, including other properties of drifting / blowing snow at the PE station.

*44. Page 15, line 8: "the occurrence of accumulation / ablation at very low wind speeds (Fig. 7). This points out that the time since the last snowfall event and the amount of low-density fresh snow that is available is of much higher importance than the wind speed in order to attain for blowing snow" => this conclusion is intriguing if we consider that friction velocity has to exceed a threshold*

*friction velocity in order to allow wind drift occurrence. If the wind speed is too low, snow saltation is not possible (e.g. Nishimura and Nemoto, 2005). Please discuss this point considering current knowledge on blowing snow processes (e.g. Nishimura and Nemoto, 2005).*

Snowpack properties mainly determine snow erosion: dendricity, density, sphericity and particles size regulate the availability of snow for transportation. These parameters change with metamorphism and impact the threshold friction velocity and thus the minimum wind speed required for particles to lift from the ground (Gallee et al., 2001; Gossart et al., 2017). As postulated by Mahesh et al. (2003), the end of a large snow storm with high wind speeds could still hold snow particles suspended in the air, even if the wind speed has already dropped to lower speeds than those required to dislodge the particles from the ground at the onset of the blowing snow event. Conversely, if no particles are available for the wind to pick up, blowing snow might not occur even though the wind speeds are high. The blowing snow threshold is therefore highly variable and depends on meteorological conditions, among others (e.g. Li and Pomeroy, 1997). Gossart et al. (2017) note that the presence of freshly fallen snow has a great impact on blowing snow occurrence and blowing snow layer height. The majority of blowing snow events occur during or within a day after a precipitation event (nearly 60 and over 80 % of the blowing snow occurrences, respectively). This complies with the theory of Mahesh et al. (2003).

*Thresholds for drifting snow initiation are very variable and depend on both the meteorological conditions and the snow particles characteristics (Li and Pomeroy, 1997; Bintanja and Reijmer, 2001).*

*An analysis of the wind roses indicates the occurrence of accumulation and ablation at very low wind speeds (Fig. 7b). This points out that the time since the last snowfall event and the availability of low-density fresh snow is of much higher importance than the wind speed in order to attain for blowing snow, confirming the results of Gallee et al. (2001); Mahesh et al. (2003); Scarchilli et al. (2010); Amory et al. (2017); Gossart et al. (2017). The time since the last snowfall event might also be an important parameter to explain the large variability in drifting snow wind speed thresholds and threshold friction velocities (Li and Pomeroy, 1997; Trouvilliez et al., 2014).*

*This freshly fallen snow has a low density and is easily redistributed as the wind speed threshold for drifting snow erosion is low (Li and Pomeroy, 1997; Mahesh et al., 2003; Trouvilliez et al., 2014).*

*45. Page 15, lines 15-20: "A total of 1125 cm in SMB changes was measured by the AWS during the events. The ceilometer was only able to detect 274 cm, indicating that the blowing snow layer only has a limited vertical extent and that the transport of snow is restricted to shallow layers close to the ground during these type of events." => I suppose that blowing snow events with a limited vertical extent are transporting weak amounts of snow. How can this type of events explain the difference between 1125 cm and 274 cm?*

The ceilometer is able to detect blowing snow layers with a height exceeding 30 meters. In literature, it is stated that blowing snow layers of greater height have the potential to transport more snow at the surface (e.g. Mahesh et al., 2003; Nishimura and Nemoto, 2005; Naaim-Bouvet et al., 2010). However, based on the net amount of snow transported during accumulation and ablation events, no difference is observed at the PE station. 851 cm (1125 cm - 274 cm) of

snow height changes at the PE station took place when the blowing / drifting snow layer was less than 30 meters in altitude. In total there are 159 events attaining for accumulation and ablation during non-snowfall conditions (SMB +, S 0 & SMB -, S 0). During 29 of them, blowing snow was detected by the ceilometer (274 cm / 29 = 9.4 cm on average), while during 95 the blowing snow was limited to less than 30 meters in altitude (851 cm / 95 = 8.9 cm on average). Due to the 24 hour moving average over the AWS height measurements, transport (and height changes) during the coarse of the event is smoothed out and only the net change in mass is taken into account.

During an accumulation event, snow is transported tens of kilometers towards the PE station (see e.g. general comment 1). However, when the event is strong (i.e. high wind speeds and a large vertical extent), part of the snow that is transported towards the PE station might also be removed again. Due to the smoothing of the AWS snow height measurements, height changes during the course of an event are not captured by our AWS and only the net total change between the start and the end of the event is recorded. In case the acoustic height sensor was measuring on a minute time resolution, more height changes would be measured during stronger blowing snow events.

During an ablation event, fresh snow is removed from the PE station. Here, the fresh snow has a low threshold wind speed and snow is easily removed up to the point that the density of the snow surface is too high for the wind to erode the surface. For ablation events, there is no large dependence on the strength (or height) of the blowing snow event.

**46. Furthermore, it is noted that "ablation and accumulation can significantly compensate each other" => please discuss this paragraph considering Libois et al. (2014) results.**

This comment refers to the hourly changes in snow height caused by sastrugi movement as discussed in Libois et al. (2014). However, in our study, changes in snow height due to local sastrugi movement are mainly erased due to the application of a 24 hour running mean on the snow height measurements of the AWS (see general comment 1).

**47. Page 15, line 10: please cite Amory et al. (2017).**

The reference has been added.

*This points out that the time since the last snowfall event and the availability of low-density fresh snow is of much higher importance than the wind speed in order to attain for blowing snow, confirming the results of Gallee et al. (2001); Mahesh et al. (2003); Scarchilli et al. (2010); Amory et al. (2017); Gossart et al. (2017).*

**48. Page 15, line 25: please cite Nishimura and Ishimaru (2012); Trouvilliez et al. (2014).**

The references have been added. Furthermore, Libois et al. (2014) was added.

*As such, in order to get a good idea of accumulation and ablation during non-precipitating time periods and its influence on the local SMB, near-ground observations of drifting / blowing snow are indispensable (Takahashi, 1985; Nishimura and Nemoto, 2005; Bellot et al., 2011; Leonard and Maksym, 2011; Leonard et al., 2012; Nishimura and Ishimaru, 2012; Barral et al., 2014; Libois et al., 2014; Trouvilliez et al., 2014).*

*49. Page 15: the first paragraph of the conclusion is not really necessary (repetition of the introduction).*

We condensed the first paragraph of the conclusion to a discussing of the methodology and data.

*In this study, snowfall and associated surface mass balance (SMB) changes (ablation / accumulation) are investigated with regard to large-scale atmospheric circulation patterns and snow erosion by wind at the Princess Elisabeth (PE) station in East Antarctica. Using a unique set of remote sensing instruments, such as a Micro Rain Radar and an Automatic Weather Station, which operated concurrent for a period of 37 months, statistical relationships between meteorology and snow erosion are obtained.*

*50. Page 16: "Meteorological conditions during snowfall, accumulation and ablation, were indicated, including their impact on the local SMB, which was largely unknown up to now." => Gorodetskaya et al. (2015) already proposed estimates of each SMB component. Please clarify the differences between both estimates, and justify why the present estimate is more accurate than the one proposed by Gorodetskaya et al. (2015).*

This paragraph has been removed. The conclusion is rewritten in order to focus more on the new results compared to the paper by Gorodetskaya et al. (2015).

*51. In the references: (Leonard et al., 2011) => (Leonard et al., 2012) Please include missing information in (Stohl et al., 1995).*

The references have been updated.

**References**

[revised manuscript text omitted]

---

## Author Response (AR2)

**Response to final comments:**
**How does the ice sheet surface mass balance relate to snowfall? Insights from a ground-based precipitation radar in East Antarctica**

Niels Souverijns, Alexandra Gossart, Irina V. Gorodetskaya, Stef Lhermitte,
Alexander Mangold, Quentin Laffineur, Andy Delcloo, Nicole P.M. van Lipzig

May 30, 2018

For clarifying our answers to the reviewers' comments, the following color scheme is used: comments of the reviewer are denoted in blue, our answers are denoted in black and quotes from the revised text are in green.

**1 Editor's comments**

*1. Note that the Vaughan et al. (1999) dataset does not just include stake observations, but snowpits and ice cores, beta-activity horizons etc. Suggest replacing "stake measurements registering only the total change in snow height" with "stake and other measurements that register only the total change in snow height".*

The text has been adapted accordingly.

*As such, most studies rely on stake and other measurements that register only the total change in snow height.*

*2. I suggest removing "/" notation in the following for clarity. "In this study we focus on the relation / interactions between snowfall and accumulation in order to understand / assess the local SMB components in more detail." This notation appears very often in the manuscript, but it would be best to minimise its use as much as possible.*

We adapted the manuscript and removed the "/" notation.

*In this study we focus on the relation and interactions between snowfall and accumulation in order to understand and assess the local SMB components in more detail.*

*... wind-induced accumulation or ablation by drifting and blowing snow ($ER_{ds}$).*

*In this study, the focus is mainly on the relation between accumulation, ablation and snowfall.*

*An event is confined to the period between the start of snowfall, accumulation or ablation and the moment no more snowfall, accumulation or ablation is observed.*

*In order to understand the mechanisms attaining for snowfall and wind-induced accumulation or ablation,...*

*In order to validate the spatial scale of accumulation and ablation,...*

*... wind-induced accumulation and ablation ...*

*As a conclusion, when the cyclone or trough is more developed,...*

*3. Change "Furthermore, a constrain on the uncertainty" to "Furthermore, a constraint on the uncertainty".*

The text has been adapted accordingly.

*4. Change "snowfall / accumulation / ablation" to "snowfall, accumulation or ablation" or perhaps "snowfall or accumulation/ablation".*

The text has been adapted accordingly.

*An event is confined to the period between the start of snowfall, accumulation or ablation and the moment no more snowfall, accumulation or ablation is observed.*

*5. Change "parametrised by a self-constructed index" to "parametrised by an index we construct".*

The text has been adapted accordingly.

*6. Change "apart that snowfall events of a longer duration have a higher chance to attain for accumulation" to "except that snowfall events of a longer duration have a higher chance to attain accumulation".*

The text has been adapted accordingly.

*7. The wording "attain for" is used many times, but this is not very commonly used in English and could perhaps cause some confusion. Often alternatives "produce", "generate", "result in" or "account for" might be more appropriate. Sometimes "attain" rather than "attain for" would be better.*

Sentences containing "attain for" have been modified.

*Generally, snow storms of longer duration and larger spatial extent have a higher chance to attain accumulation at the local scale, while shorter events usually result in ablation (on average 17 and 12 hours respectively).*

*Sensitivity studies have been executed on this threshold (varying between 80 and 100 %) and account for only small relative differences not influencing the general conclusions significantly.*

*As such, the larger the cloud system or the duration of the event, the higher the chance that snowfall events result in accumulation at a local scale, rather than ablation.*

*Almost all areas north of the mountain ridge generate snow accumulation in the RACMO2.3 simulation when there is accumulation but no snowfall at the PE station.*

*This points out that the time since the last snowfall event and the availability of low-density fresh snow is of much higher importance than the wind speed in order to generate blowing snow,...*

*Longer and larger events result in bigger areas with fresh snow deposition, allowing snow from synoptically upstream areas to be transported towards the PE stations.*

**2   Reviewer 1**

*1. Fig. 5 caption: just to be clearer, I would add "top panel" and "bottom panel" talking about the red dot and line. "The red dot (top panel) denotes the location of the Princess Elisabeth station, while the line (bottom panel) shows the time the snowfall event arrived at the station".*

The caption has been adapted accordingly.

**3   Reviewer 2**

*1. In their response, the authors write that in the analysis of Libois et al. (2014), sastrugis are associated to high peaks in snow height measurements lasting only a few hours, which should disappear if a running mean of 24 hours is applied on the hourly snow height measurements. =¿ if the authors look more accurately to Figure 2 from Libois et al. (2014), with a focus on surface height variations in august/September 2012, they will observe that the sonic gauge proposes 1) a significant snow height increase, 2) constant values during a few day 3) a significant decrease of the surface (erosion). These events are probably due to the movement of sastrugis, which likely stopped moving below the sonic gauge during a few day and then was eroded again. The authors acknowledge that 3 reptation events may impact their results but conclude that: "Apart from this, temporary peaks (i.e. with a strong decay within 48 hours) in the snow height records are excluded from the analysis". I suggest that the authors smooth their comment and write that a 24 hour running mean should remove the "majority" of peaks associated to the motion of sastrugis. Nevertheless, few events may also be related to artefacts caused by the motion of sastrugis.*

This is a correct remark of the reviewer. As noted, this issue was acknowledged in the revised manuscript. The rephrasing proposed here clarifies how sastrugis are dealt with and is added to the manuscript.

*A running mean of 24 hours is applied to erase the majority of the short-term decameter scale variability due to sastrugi movement (Libois et al., 2014). Nevertheless, few events may still be related to artefacts caused by the motion of sastrugis.*

*2. the authors write that "Since the magnitude of the two sublimation terms is small over the PE station, the uncertainty on the ERds component is mainly determined by the uncertainty of the snowfall estimate". However, the cumulative Sublimation and ERds values are in the same range of magnitude. As a consequence, significant uncertainty in Sublimation can impact significantly the ERds values. Many papers discuss the uncertainty of surface sublimation (e.g. Bliss et al. (2011)). Getting the uncertainty of the SUds term is more complex but may be discussed using results from RACMO2.3 model => I suggest that the authors indicate an uncertainty range for both sublimation terms, or give values (or statistics) of each term for several typical events demonstrating that ERds is determined by snowfall estimate.*

The uncertainty on both sublimation terms has been calculated before by Gorodetskaya et al. (2015) for the Princess Elisabeth station. They attained uncertainty values of 47 % and 20 % for the surface and blowing snow sublimation term respectively. This uncertainty range was stated in Section 2.1. Compared to the uncertainty on the snowfall term (approximately 60 %), their influence is less important. Also, the snowfall term has a larger contribution to the uncertainty of ERds due to its higher absolute magnitude. We added a sentence to the text referring to the lower uncertainty ranges of both sublimation terms.

[revised manuscript text omitted]